# δ-Catenin controls astrocyte morphogenesis via layer-specific astrocyte–neuron cadherin interactions

Christabel Xin Tan[1], Dhanesh Sivadasan Bindu[1], Evelyn J. Hardin[1], Kristina Sakers[1], Ryan Baumert[1], Juan J. Ramirez[2], Justin T. Savage[2], and Cagla Eroglu[1,2,3]

**Astrocytes control the formation of specific synaptic circuits via cell adhesion and secreted molecules. Astrocyte synaptogenic functions are dependent on the establishment of their complex morphology. However, it is unknown if distinct neuronal cues differentially regulate astrocyte morphogenesis. δ-Catenin was previously thought to be a neuron-specific protein that regulates dendrite morphology. We found δ-catenin is also highly expressed by astrocytes and required both in astrocytes and neurons for astrocyte morphogenesis. δ-Catenin is hypothesized to mediate transcellular interactions through the cadherin family of cell adhesion proteins. We used structural modeling and biochemical analyses to reveal that δ-catenin interacts with the N-cadherin juxtamembrane domain to promote N-cadherin surface expression. An autism-linked δ-catenin point mutation impaired N-cadherin cell surface expression and reduced astrocyte complexity. In the developing mouse cortex, only lower-layer cortical neurons express N-cadherin. Remarkably, when we silenced astrocytic N-cadherin throughout the cortex, only lower-layer astrocyte morphology was disrupted. These findings show that δ-catenin controls astrocyte–neuron cadherin interactions that regulate layer-specific astrocyte morphogenesis.**

## Introduction

The cerebral cortex is central to sensory input processing and coordination of complex behaviors. Cortical development is intricate, yet the underlying cellular processes are analogous between the human and murine brain (Molnár et al., 2019). As illustrated in Fig. 1 a, radial glial cells are neural progenitor cells found in the subventricular zone that first give rise to neurons and, later, glial cells such as astrocytes (Qian et al., 2000; Urbán and Guillemot, 2014). In mice, cortical neurogenesis peaks between embryonic day (E)13–16 and neurons are born in an "inside-out" manner where lower-layer neurons are born first (Shen et al., 2006). The switch to astrogenesis occurs approximately at E18 and most cortical astrocytes are born postnatally (Miller and Gauthier, 2007). Newly born astrocytes migrate along radial glial cell processes to different cortical layers and undergo local division (Ge and Jia, 2016). Between postnatal day (P)7 and 28, astrocytes undergo a period of morphological maturation: primary processes radiating out from the cell soma branch out extensively into secondary, tertiary, and fine peri-synaptic astrocyte processes, resulting in the elaborate "star"-like morphology that astrocytes are named for (Bushong et al., 2004; Morel et al., 2014; Torres-Ceja and Olsen, 2022).

Cortical synaptogenesis peaks concurrently with astrocyte morphological maturation (Rice and Barone, 2000; Li et al., 2010; Stogsdill et al., 2017), highlighting the necessity of bidirectional signaling between astrocytes and neurons during cortical development. Astrocytes directly regulate synapse formation, maturation, and function of specific neuronal circuits through secretory cues and cell adhesion molecules (Chung et al., 2015; Farhy-Tselnicker and Allen, 2018; Tan et al., 2021). Neuronal signaling is equally instructive for astrocyte development. Inversion of neuronal cortical layers in *Reeler* mice reverses the expression of upper- and lower-layer cortical astrocyte markers (Lanjakornsiripan et al., 2018; Bayraktar et al., 2020). Notably, cortical astrocytes use neuronal adhesion to achieve morphological complexity (Stogsdill et al., 2017).

Once considered a homogenous population, adult cortical astrocytes are now recognized as heterogenous in their gene expression, morphology, and function (Takata and Hirase, 2008; Lanjakornsiripan et al., 2018; Miller, 2018; Bayraktar et al., 2020). Because cortical astrocyte heterogeneity could be attributed to cortical layer specialization, it has been hypothesized that layer-specific neuron heterogeneity could directly or indirectly regulate cortical astrocyte heterogeneity (Lanjakornsiripan et al.,

[1]Department of Cell Biology, Duke University School of Medicine, Durham, NC, USA; [2]Department of Neurobiology, Duke University School of Medicine, Durham, NC, USA; [3]Howard Hughes Medical Institute, Duke University School of Medicine, Durham, NC, USA.

Correspondence to Cagla Eroglu: cagla.eroglu@duke.edu.

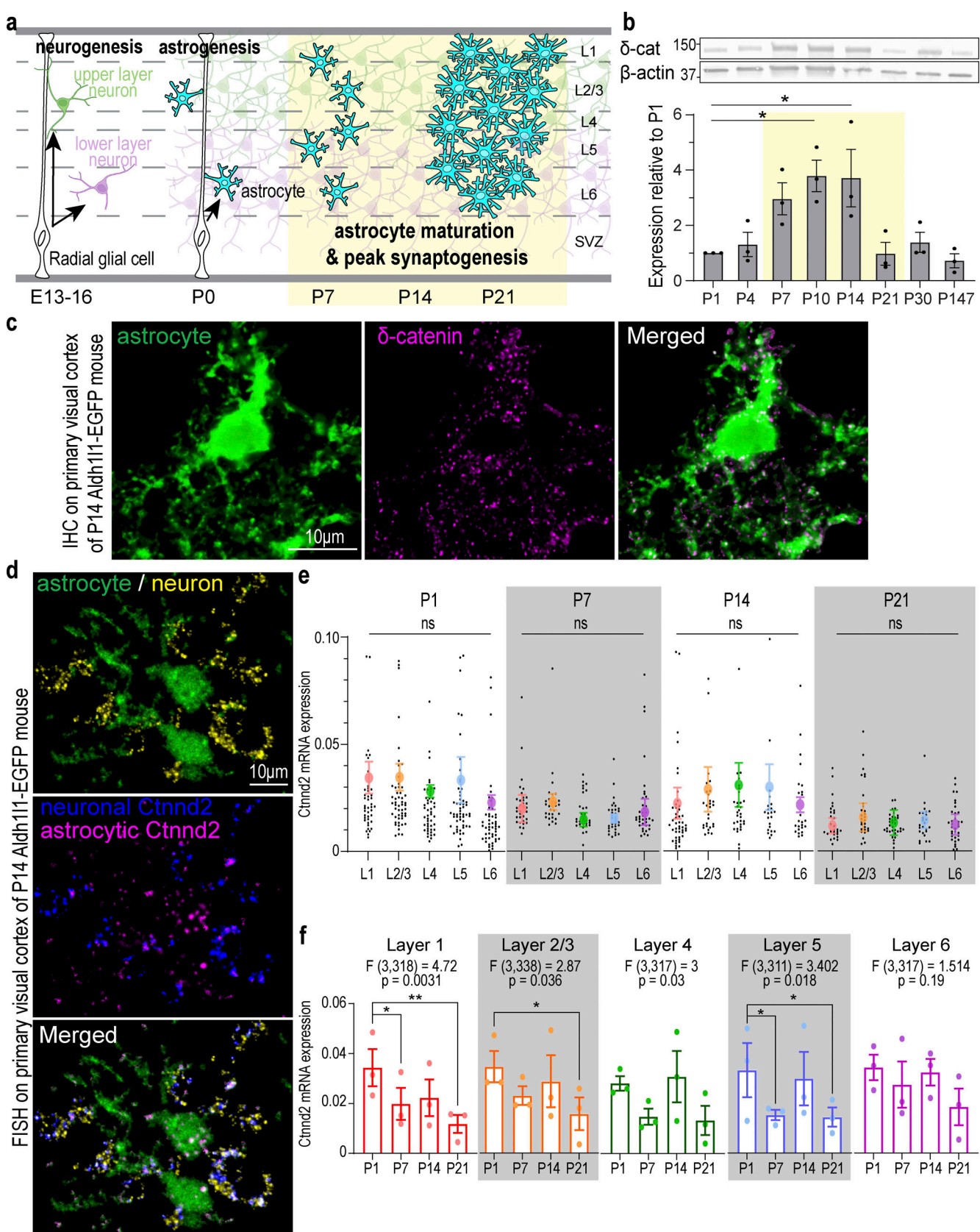

Figure 1. **Cortical astrocytes and neurons express Ctnnd2/δ-catenin during early postnatal development. (a)** Schematic of neurogenesis, astrogenesis, and astrocyte maturation in the developing murine cortex. Astrocyte morphological maturation is concurrent with peak synapse formation in the second postnatal week. **(b)** Top: Western blot analysis of δ-catenin (δ-cat) protein expression WT mouse cortex from P1 to adult. β-Actin: loading control. All protein

molecular weights are expressed in kiloDaltons (kD). Bottom: δ-Catenin expression peaks at P10–14 (P = 0.011 and P = 0.013, respectively). $n$ = 3 mice/time point. One-way ANOVA, Dunnett's post-test. **(c)** Detection of *Ctnnd2* mRNA in cortical astrocytes and neurons. RNA probes specific to βIII-tubulin (neuronal marker) labeled with Opal 690 and Ctnnd2 labeled with Opal 570 were hybridized to 16 µm V1 cortical sections of Aldh1l1-EGFP P14 mice. Neuronal (pseudo-colored magenta) and astrocytic Ctnnd2 (pseudo-colored blue) were identified by colocalization with astrocytic and neuronal markers on Imaris software. **(d)** δ-Catenin (magenta) in cortical astrocyte of Aldh1l1-eGFP mice at P14. **(e)** Ctnnd2 mRNA is expressed at comparable levels in astrocytes across all cortical layers at P1 (P = 0.78), P7 (P = 0.75), P14 (0.93), and P21 (P = 0.98). Quantification of mRNA expression by normalizing the volume of Ctnnd2 mRNA puncta to volume of astrocyte soma. Astrocytes from all layers of the V1 cortex (L1 [red], L2/3 [orange], L4 [green], L5 [blue], and L6 [purple]) were imaged and analyzed. Individual astrocytes are plotted in black while the mean and SEM of each animal are plotted in color. $n$ = 83 astrocytes from three animals per layer per timepoint. Nested one-way ANOVA for each time point. **(f)** Ctnnd2 mRNA expression L1 (red), L2/3 (orange), L4 (green), L5 (blue), and L6 (purple) is significantly altered during the first 3 wk of cortical development. Raw data is identical to e, but analyzed using developmental time point as the variable using two-way ANOVA with Dunnett's multiple comparison's test. In L1 astrocytes, Ctnnd2 mRNA expression is significantly decreased in P7 (P = 0.023) and P21 (P = 0.001) compared with P1. In L2/3 astrocytes, Ctnnd2 mRNA expression is significantly decreased in P21 relative to P1 (P = 0.019). In L5 astrocytes, Ctnnd2 mRNA is significantly decreased in P7 (P = 0.028) and P21 (P = 0.046) compared to P1. Each dot on the bar graph represents the average Ctnnd2 mRNA expression from each animal. All error bars represent SEM. Scale bars: 10 µm. * P < 0.05; ** P < 0.01. Source data are available for this figure: SourceData F1.

2018; Bayraktar et al., 2020). However, it is unknown if cortical astrocyte heterogeneity is already specified during development and what the molecular mechanisms underlying neuron-mediated layer-specific astrocyte heterogeneity are.

Here, we report that δ-catenin (gene name: *Ctnnd2*), previously thought to be neuron-specific, is also expressed in astrocytes and regulates neuron-contact-dependent cortical astrocyte morphogenesis. δ-Catenin was first identified as a neural-specific member of the p120-catenin subfamily of proteins (Paffenholz and Franke, 1997). Although *Ctnnd2* mRNA transcript was detected in rat glia cultures (Kawamura et al., 1999), subsequent characterization of δ-catenin in the embryonic and developing murine brain suggested δ-catenin to be restricted to the neuronal lineage (Kosik et al., 2005). δ-Catenin is first expressed as an adherens junction protein between neighboring neuronal progenitor cells, transiently down-regulated during the shift from neuronal proliferation to neuronal migration, and then primarily expressed in dendrites of postmitotic neurons (Ho et al., 2000). Since then, δ-catenin has been studied mainly as a critical regulator of dendrite morphology through its modulation of Rho-GTPase activity and interactions with PDZ proteins (Martinez et al., 2003; Arikkath et al., 2008; Abu-Elneel et al., 2008; Matter et al., 2009; Baumert et al., 2020; Donta et al., 2022).

δ-Catenin is linked to numerous neurodevelopmental and neuropsychiatric disorders. Copy number variations and mutations in *Ctnnd2*, identified through genome-wide association studies and patient case studies, have been linked to autism spectrum disorder (Turner et al., 2015; Miller et al., 2020), intellectual disability (Hofmeister et al., 2015; Belcaro et al., 2015), Cri-du-chat syndrome (Sardina et al., 2014; Medina et al., 2000), attention deficit and hyperactivity disorder (Adegbola et al., 2020), and schizophrenia (Vrijenhoek et al., 2008; Wang et al., 2010; Nivard et al., 2014). δ-Catenin N-terminal mice, expressing a mutant form of δ-catenin that lacks its central armadillo and C-terminal domains, also possessed cognitive deficits, demonstrating severe deficiencies in spatial learning compared with wild-type controls (Israely et al., 2004; Yuan et al., 2015).

Because it has been assumed that δ-catenin's role in the brain is neuron-specific, research into δ-catenin and how it contributes to the pathophysiology of neurological disorders has mainly been studied in neurons (Israely et al., 2004; Kosik et al., 2005; Abu-Elneel et al., 2008; Arikkath et al., 2009; Turner et al., 2015;

Yuan et al., 2015; Seong et al., 2015; Baumert et al., 2020). However, data from cell-type-specific and single-cell transcriptomic analyses have shown that *Ctnnd2* mRNA is also highly expressed in astrocytes both in mice (Zhang et al., 2014; Farhy-Tselnicker et al., 2021) and humans (Zhang et al., 2016; Nowakowski et al., 2017). Therefore, we set out to understand if δ-catenin is expressed in astrocytes and what the role of astrocytic δ-catenin in the developing brain might be. Using purified astrocyte–neuron co-culture, structural modeling, and biochemical analyses, we found that both astrocytic and neuronal δ-catenin are necessary for astrocyte morphogenesis through its interaction with and control of cadherin cell surface localization. Further, through astrocyte labeling in vivo and reconstructing and quantifying astrocyte morphologies, we show that this critical astrocyte–neuron adhesion system regulates layer-specific astrocyte morphogenesis. Our findings reveal a molecular mechanism underlying how neuronal adhesion establishes cortical astrocyte heterogeneity.

## Results

### Cortical astrocytes and neurons express *Ctnnd2*/δ-catenin during early postnatal development

δ-Catenin expression is temporally dynamic during brain development (Ho et al., 2000). However, because most developmental δ-catenin studies were conducted using hippocampal cultures (Abu-Elneel et al., 2008; Arikkath et al., 2008, 2009; Yuan et al., 2015; Baumert et al., 2020), δ-catenin expression patterns during key periods of postnatal cortical development are unknown. We observed via Western blot of mouse cortical lysates that δ-catenin expression peaks between P10 and 14 (Fig. 1 b). This period is of particular significance as this is the window during cortical development when synapse formation and astrocyte morphogenesis occur concurrently (Stogsdill et al., 2017; Farhy-Tselnicker and Allen, 2018; Takano et al., 2020; Baldwin et al., 2021).

Next, we asked whether cortical astrocytes express δ-catenin. δ-Catenin is expressed in hippocampal and cortical dendrites and is required for proper dendritic morphology and spine maturity (Martinez et al., 2003; Abu-Elneel et al., 2008; Arikkath et al., 2008, 2009; Kim et al., 2008; Yuan et al., 2015; Baumert et al., 2020). However, various transcriptomic studies have challenged the assumption that δ-catenin expression is

neuron-specific. Mouse *Ctnnd2* mRNA transcript level is 1.98-fold higher in P7 astrocytes relative to neurons (Fig. S1 a; Zhang et al., 2014), and *Ctnnd2* mRNA is actively transcribed in adult murine astrocytes (Fig. S1 b; Srinivasan et al., 2016). These observations are echoed in the developing human brain. *CTNND2* mRNA is expressed in both fetal astrocytes and mature human astrocytes (Fig. S1 c; Zhang et al., 2016), while single-cell transcriptomic analysis of the human telencephalon revealed that *CTNND2* is most abundant in astrocytes and excitatory neurons in the primary visual cortex (V1) and prefrontal cortex (Fig. S1 d; Nowakowski et al., 2017). Therefore, we wondered if δ-catenin is expressed by astrocytes and whether δ-catenin participates in cortical astrocyte development. Indeed, we detected robust δ-catenin expression within P14 cortical astrocyte soma and processes within the mouse V1 visual cortex (Fig. 1 c).

Next, we verified that *Ctnnd2* mRNA is present in both mouse cortical astrocytes and neurons in vivo by multiplex RNA fluorescence in situ hybridization (Fig. 1 d). We then quantified astrocytic *Ctnnd2* mRNA expression by normalizing the volume of *Ctnnd2* mRNA puncta with the astrocyte soma (labeled by EGFP) across different layers (L1, L2/3, L4, L5, and L6) of the mouse V1 visual cortex at P1, P7, P14, and P21. We found that astrocytic *Ctnnd2* mRNA levels did not vary significantly across cortical layers (Fig. 1 e). However, comparing the *Ctnnd2* mRNA expression of each layer across the first three postnatal weeks revealed that astrocytic *Ctnnd2* mRNA expression in layers 1, 2/3, 4, and 5 are significantly different across this critical developmental period (Fig. 1 f). These RNA–fluorescence in situ hybridization (FISH) experiments also revealed that astrocytic *Ctnnd2* mRNA expression decreases significantly by P21, congruent with our previous quantification of δ-catenin expression in vivo (Fig. 1 b).

Next, we compared *Ctnnd2* mRNA and δ-catenin protein abundance in primary cortical astrocyte and neuron cultures (Fig. S1 e). Astrocytes and neurons were isolated separately from the P1 rat cortex and cultured independently for 9 d in vitro (DIV). The resulting astrocyte cultures did not contain detectable neuronal contamination, as evidenced by extremely low levels of neuron-specific enolase (NSE, Fig. S1 f) and the absence of βIII-tubulin protein (Tuj1, Fig. S1 g). Quantification of glial fibrillary acid mRNA and protein suggested that neuronal cultures contain some astroglia contamination (*Gfap*, Fig. S1, f and g). We found by quantitative RT-PCR (RT-qPCR) that cortical astrocytes and neurons express comparable levels of *Ctnnd2* (Fig. S1 f). Similarly, δ-catenin protein was also detected via Western blot both in primary neuron and astrocyte cultures (Fig. S1 g). These results show that cortical astrocytes express *Ctnnd2* mRNA and δ-catenin protein at comparable levels to neurons in vitro and in vivo.

### Astrocytic *Ctnnd2*/δ-catenin is required for neuron contact-mediated astrocyte morphogenesis in vitro

Having established that δ-catenin is also expressed in astrocytes and that its expression in the cortex peaks during the critical window of astrocyte morphological maturation, we investigated the function of δ-catenin in astrocyte morphogenesis. First, we used an established cortical astrocyte–neuron co-culture assay

(Fig. 2 a; Stogsdill et al., 2017; Baldwin et al., 2021) to test the necessity of Ctnnd2 expression in astrocytes for neuronal contact–induced morphogenesis. To do so, we silenced *Ctnnd2* expression using short hairpin RNA (shRNA) specifically in wild-type rat astrocytes and assessed the morphology of these transfected astrocytes after 48 h co-culture with wild-type neurons. Wild-type astrocytes in vitro only gain a complex morphology when cultured on top of neurons (Stogsdill et al., 2017). We hypothesized that knocking down *Ctnnd2* would reduce astrocyte complexity if δ-catenin is required for astrocyte morphogenesis.

Two shRNAs (shCtnnd2-1, sh1 and shCtnnd2-2, sh2), each targeting different regions of *Ctnnd2*, were used to knock down *Ctnnd2* expression (Fig. S2 a). Astrocytes transfected with shControl, a plasmid containing a non-targeting scrambled sequence of shCtnnd2-1, have an elaborate morphology characterized by primary, secondary, and tertiary processes as expected of wild-type astrocytes co-cultured with neurons (Fig. 2 b). In contrast, *Ctnnd2* knockdown astrocytes had fewer primary processes, stunted secondary processes, and were devoid of tertiary processes (Fig. 2 b). Astrocyte branching complexity of shCtnnd2-1 and shCtnnd2-2 knockdown astrocytes was significantly reduced when quantified by Sholl analysis (Fig. 2 c). To determine if δ-catenin is required for astrocyte morphogenesis independent of neuronal contact, we also measured astrocyte morphology in an astrocyte monoculture system (Fig. 2 a). At DIV 11, transfected astrocytes were co-cultured onto a layer of wild-type astrocytes instead of wild-type neurons (as in the co-culture system) for 48 h in serum-free media. As expected, both shControl and shCtnnd2 astrocytes in the astrocyte monoculture were simple, as evidenced by the significantly reduced complexity in astrocyte processes when compared with shControl astrocytes co-cultured with neurons (P < 2.2 × 10$^{-16}$ both conditions, Fig. 2, b and c). In the absence of neurons, shControl and shCtnnd2 monoculture astrocytes were indistinguishable from each other in terms of morphology (P = 0.09, Fig. 2, b and c). Both shCtnnd2 monoculture astrocytes and shCtnnd2 astrocytes co-cultured with neurons had similar complexities (P = 0.97) when quantified by Sholl analysis (Fig. 2, b and c), giving us confidence that astrocytic δ-catenin is required for neuron contact-mediated astrocyte morphogenesis.

δ-Catenin was previously shown to control neuronal dendrite complexity (Kim et al., 2008; Abu-Elneel et al., 2008; Arikkath et al., 2008; Baumert et al., 2020). Thus, we wondered whether astrocytic Ctnnd2 could impact neuronal dendrite morphogenesis. To do so, we quantified neuronal morphology in a similar co-culture assay (Fig. S2 b), in which wild-type neurons were plated onto either shCtnnd2 or shControl transfected astrocytes. We found that silencing astrocytic Ctnnd2 does not impact neuronal morphology in a non-cell autonomous manner (Fig. 2, c and d).

Because astrocytes also impact each other's morphology, we next tested if astrocyte–astrocyte interactions are altered when Ctnnd2 is knocked down. To do so, we utilized astrocyte mosaic cultures (Fig. S2 e). Briefly, shControl-GFP or shCtnnd2-GFP nucleoporated astrocytes were co-cultured with shControl-mCherry or shCtnnd2-mCherry nucleoporated astrocytes in a

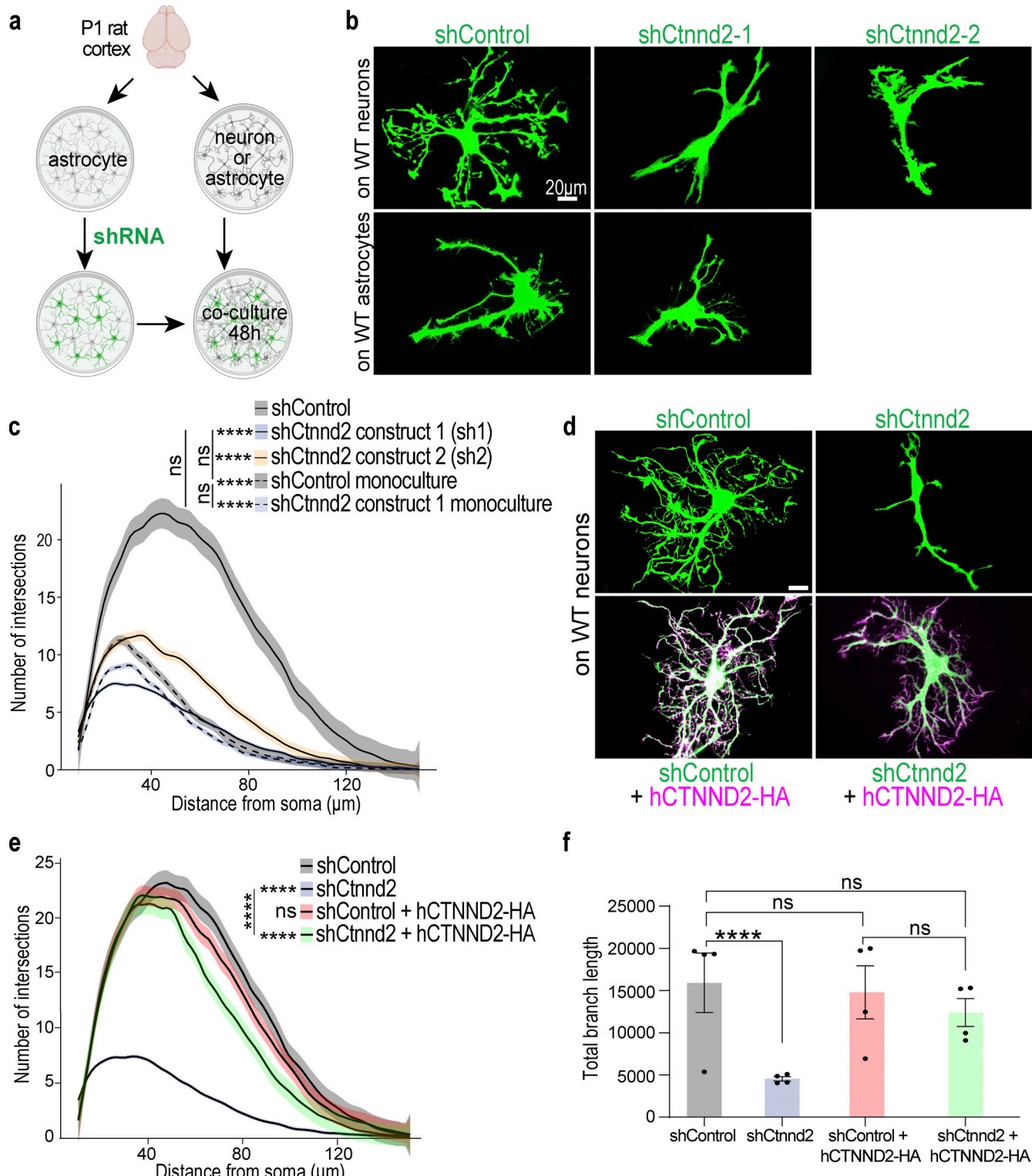

**Figure 2. Astrocytic *Ctnnd2*/δ-catenin is required for neuron contact-mediated astrocyte morphogenesis in vitro. (a)** Schematic of astrocyte–neuron co-culture and astrocyte monoculture assay. Astrocytes and neurons are independently isolated from P1 rat cortices and cultured separately. DIV 9 astrocytes are transfected with pLKO.1 plasmid expressing hU6-shRNA and CAG-EGFP. Transfected astrocytes are co-cultured on top of wild-type neurons (co-culture) or wild-type astrocytes (monoculture) at DIV 11 for 48 h. **(b)** Representative images of rat astrocytes transfected with control shRNA (shControl), shCtnnd2-1, or shCtnnd2-2 (EGFP positive) after 48 h co-culture with wild-type neurons or wild-type astrocytes (not labeled). **(c)** Silencing δ-catenin expression using shCtnnd2-1 (P < 2.2 × 10⁻¹⁶) or shCtnnd2-2 (P < 2.2 × 10⁻¹⁶) significantly reduced astrocyte complexity when co-cultured on top of neurons. Silencing δ-catenin expression using shCtnnd2-1 did not alter astrocyte morphology when cultured with wild-type astrocytes (P = 0.09). Astrocyte complexity is quantified by Sholl analysis. *n* = 85–101 astrocytes per condition from three independent co-culture experiments for astrocyte–neuron co-culture assay and astrocyte monoculture assay. Linear mixed model with Tukey HSD. **(d)** Representative images of shRNA expressing astrocytes (EGFP/green) in the presence or absence

of full-length human CTNND2 (hCTNND2) construct with a C-terminal HA tag (magenta) after 48 h co-culture with wild-type neurons (not labeled). **(e)** Complexity of shCtnnd2 astrocytes is restored by shRNA-resistant, full-length human δ-catenin protein (hCTNND2-HA, $P < 2.2 \times 10^{-16}$). $n$ = 106–128 astrocytes per condition from four independent experiments, linear mixed model with Tukey HSD. ns, not significant. **(f)** Total astrocyte branch length of shCtnnd2 astrocytes is significantly reduced compared to shControl, shControl + hCTNND2-HA, and shCtnnd2 + hCTNND2-HA astrocytes (**** $P < 0.0001$ for each pairwise comparison). $n$ = 97–110 astrocytes per condition from four independent experiments. Nested one-way ANOVA, Tukey HSD. ns, not significant. All data are presented as mean ± SEM. Scale bars: 20 μm.

1:9 ratio for 48 h in serum-free media. Nucleoporation was utilized as it afforded higher transfection efficiency, and only the sparse GFP-positive astrocytes were assessed for morphology changes.

As expected, astrocytes in this mosaic culture paradigm were less complex than astrocytes in the neuron co-culture system, with a peak of only 8–10 intersections when quantified by Sholl analysis across all conditions tested (Fig. S2, f and g). The sparse population of GFP-labeled shControl-transfected astrocytes plated onto mCherry-labeled shControl astrocytes served as our control for this experiment. Silencing Ctnnd2 in the larger population of astrocytes did not influence the morphology of the shControl or shCtnnd2-transfected astrocytes (P = 0.18 and P = 0.06 respectively, Fig. S2, f and g). The same was true for the sparse GFP-labeled shCtnnd2-transfected astrocytes cultured with shControl or shCtnnd2 astrocytes (P = 0.91, Fig. S2, f and g). Taken together, these experiments suggest that δ-catenin knockdown specifically impacts astrocyte morphogenesis, which is induced by neuronal contact, and does not impact astrocyte morphology via alteration of astrocyte–astrocyte adhesions.

Next, we performed in vitro rescue experiments with a hemagglutinin (HA)-tagged, shRNA-resistant human δ-catenin protein (hCTNND2-HA, Fig. S2 h). Coexpression of hCtnnd2-HA with shCtnnd2-1 (henceforth named shCtnnd2) rescued astrocyte morphogenesis, as evidenced by a significant increase in astrocyte complexity reflecting the restoration of secondary branch length and presence of tertiary branches (Fig. 2, d and e). However, Sholl analysis (Fig. 2 e) also showed that shCtnnd2 + hCTNND2-HA astrocytes were statistically significantly different from shControl astrocytes (P = $3.3 \times 10^{-4}$) and shControl + hCtnnd2-HA astrocytes (P = 0.002). Because the peak number of intersections is similar across all three conditions, the difference is likely due to shCtnnd2 + hCTNND2-HA astrocytic processes not extending outward as much as shControl astrocytes (Fig. 2 d), which is, in turn, reflected as a leftward shift of the curve (Fig. 2 e). Overexpression of δ-catenin in shControl astrocytes did not further increase astrocyte complexity (P = 0.69).

Next, we analyzed total astrocyte branch length as a secondary measure of astrocyte complexity (Fig. 2 f). shControl astrocytes have a total branch length of 14,523 ± 781 μm (mean and SEM). Silencing astrocytic δ-catenin reduced total branch length by threefold (4,551 ± 228 μm), which is rescued by hCTNND2-HA (12,753 ± 595 μm). Like Sholl analysis quantification, overexpression of δ-catenin in astrocytes did not increase total branch length (13,853 ± 764 μm). Critically, the total branch length of shControl, shControl + hCTNND2-HA, and shCtnnd2 + hCTNND2-HA astrocytes were statistically the same (Fig. 2 f). Our findings show that loss of δ-catenin severely stunts

astrocyte complexity. Furthermore, δ-catenin is necessary for neuronal-contact-dependent astrocyte morphogenesis because overexpression of hCTNND2-HA is sufficient to rescue astrocyte complexity.

## δ-Catenin interacts with the juxtamembrane domain (JMD) of cadherin to control cell surface expression

How does δ-catenin regulate astrocyte morphogenesis? We hypothesized that δ-catenin mediates cadherin-dependent cell adhesion between astrocytes and neurons in the developing cortex to regulate astrocyte morphogenesis. Cadherins are a family of calcium-dependent cell adhesion molecules critical throughout neural development with crucial roles ranging from neuroblast migration, axon finding, and synapse formation (Suzuki and Takeichi, 2008; Hirano and Takeichi, 2012). Cadherins are known binding partners of δ-catenin (Silverman et al., 2007; Yuan et al., 2015), but how this interaction occurs has primarily been inferred from data for E-cadherin/p120-catenin interaction (Ishiyama et al., 2010) due to the high sequence homology between p120-catenin and δ-catenin (Paffenholz and Franke, 1997).

δ-Catenin comprises an N-terminal coiled-coil domain and a proline-rich region, a central ARM domain composed of nine armadillo repeats, and a C-terminal PDZ-binding motif (Fig. 3 a; Martinez et al., 2003; Turner et al., 2015). Cadherins, on the other hand, are single-pass transmembrane proteins that comprise five extracellular cadherin domains, a transmembrane region, an intracellular JMD, and an unstructured cytoplasmic tail (Fig. 3 b). p120-catenin ARM domain binds to cadherin JMD (Lu et al., 1999; Ounkomol et al., 2010). To specifically examine the putative interaction between δ-catenin and cadherin, we used Alphafold and PyMOL to model the interaction of δ-catenin and N-cadherin, the most abundant cadherin in the brain. As proposed by the p120/E-cadherin homology, modeling predicted that the δ-catenin ARM domain forms a solenoid structure and binds to N-cadherin JMD (Fig. 3 c). N-cadherin JMD is predicted to bind directly on a positively charged cadherin-binding groove within this solenoid structure (Fig. 3 d).

We next tested the necessity of JMD for δ-catenin/N-Cadherin interaction through co-immunoprecipitation (Co-IP) assays in HEK293T cells. The N-cadherin JMD sequence, conserved across human, mouse, and chicken (Hatta et al., 1988; Piccoli et al., 2004), was cloned out from full-length N-cadherin tagged with C-terminal GFP (N-cadherin-GFP) to obtain N-cadherinΔJMD-GFP. When co-expressed with human δ-catenin with a C-terminus HA tag (δ-catenin-HA), only full-length N-cadherin co-immunoprecipitated with δ-catenin (Fig. 3 e), experimentally validating the model where N-cadherin JMD is necessary for binding to δ-catenin.

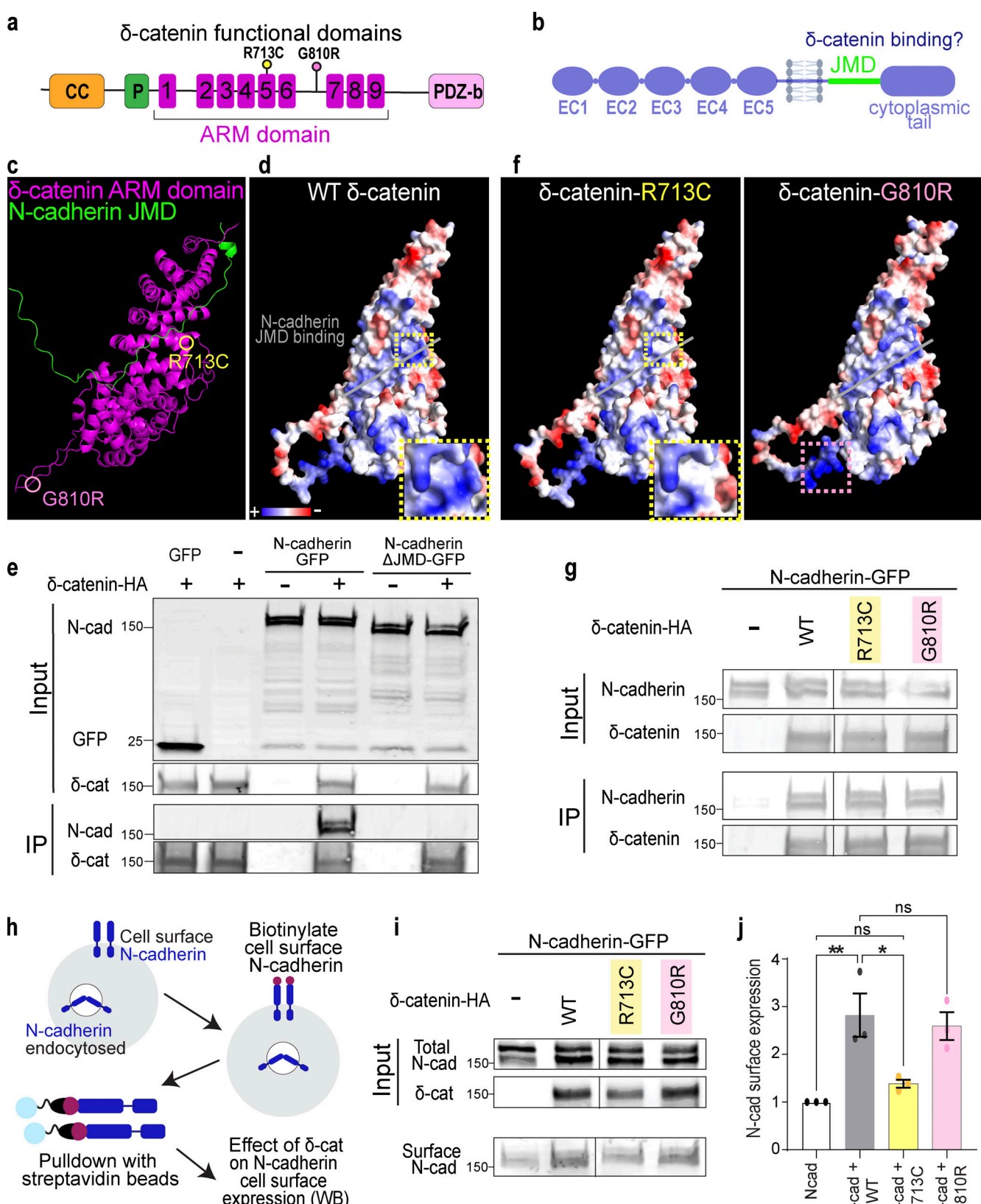

Figure 3. **δ-Catenin interacts with the juxtamembrane domain of N-cadherin to increase N-cadherin cell surface expression. (a)** Schematic of domains within δ-catenin. **(b)** Schematic of the structure of classical cadherin. The JMD of cadherin is highlighted in green. **(c)** Model of predicted interaction between N-cadherin-JMD and δ-catenin ARM domain. δ-catenin ARM domain forms a solenoid structure (magenta). N-cadherin JMD (green) binds to a groove within this solenoid structure. **(d)** Model of electrostatic charges of wild-type human δ-catenin solenoid structure. N-cadherin JMD (green) is predicted to bind to a positively charged groove within the solenoid structure. **(e)** IP of N-cadherin by δ-catenin-HA pulldown. Deletion of JMD from N-cadherin abolishes

N-Cadherin/δ-catenin interaction. Immunoblot of HEK293T cell lysates overexpressing δ-catenin-HA with GFP or full-length N-cadherin-GFP or N-cadherin without JMD-GFP. N-cadherin is detected by an anti-GFP antibody, while δ-catenin is detected by an anti-HA antibody. **(f)** Model of electrostatic charges of δ-catenin with an autism-linked point mutation R713C (left) and a polymorphism G810R (right). Both mutations are within the armadillo repeats, but only the R713C mutation (yellow dotted box) is within the groove predicted to be critical for N-cadherin JMD binding. **(g)** Immunoblot of WT, R713C, and G810R co-immunoprecipitated with full-length N-cadherin-GFP from HEK293T cell lysates. Point mutations to δ-catenin do not disrupt cadherin–catenin interaction. **(h)** Schematic of surface biotinylation experiment to determine if δ-catenin mutations alter N-cadherin cell surface expression. **(i)** Immunoblot of N-cadherin cell surface expression in HEK293T cell lysates transfected with WT, R713C, or G810R. **(j)** Expression of WT and G810 significantly increases N-cadherin expression at the cell surface (P = 0.006 and P = 0.014, respectively). Expression of R713C did not alter N-cadherin cell surface expression (P = 0.75). * P < 0.05; ** P < 0.01. n = 3 independent replicates. One-way ANOVA with Tukey HSD. All protein molecular weights are expressed in kiloDaltons (kD). All data are presented as mean ± SEM. Source data are available for this figure: SourceData F3.

Given the numerous links between δ-catenin and neurological disorders, we next modeled how two known disease-linked missense mutations, R713C and G810R, found within the δ-catenin ARM domain might alter interactions between N-cadherin and δ-catenin. R713C is a mutation previously associated with severe forms of autism (Turner et al., 2015). We predicted this mutation through protein modeling to neutralize positive charges within the cadherin-binding groove (insets in Fig. 3, d and f). G810R, on the other hand, is a polymorphism in δ-catenin associated with age-related cataracts and Alzheimer's disease (Jun et al., 2012). We did not expect this mutation to alter N-cadherin/δ-catenin interaction since this residue is outside the JMD (Fig. 3 f). However, neither R713 nor G810 was identified as residues necessary for the interaction of N-cadherin JMD with δ-catenin in our model. Thus, we postulated that N-cadherin/δ-catenin interaction would be preserved despite these mutations. The point mutations were cloned into the human δ-catenin-HA as described in Fig. 2 e and cotransfected individually with N-cadherin-GFP into HEK293T cells for Co-IP assays. N-cadherin co-immunoprecipitated with wild-type human δ-catenin (WT), δ-catenin with R713C mutation (R713C), and δ-catenin with G810R mutation (G810R) as we expected (Fig. 3 g).

Because p120-catenin regulates E-cadherin cell surface expression by inhibiting protein turnover (Davis et al., 2003; Fukumoto et al., 2008), we wondered if δ-catenin would similarly regulate N-cadherin cell surface expression and if R713C and G810R mutations might be sufficient to alter N-cadherin cell surface expression. We performed a cell surface protein biotinylation assay to isolate and quantify cell surface expression of N-cadherin in HEK293 cells transfected with WT, R713C, and G810R. Briefly, cells were placed on ice and treated with sulfonated biotin for 30 min to label cell-surface proteins before cell lysis and pulldown with streptavidin beads. Input (total) and pulldown (cell surface) lysates were subjected to SDS-PAGE separation, transferred onto the polyvinylidene difluoride membrane, and probed with antibodies (Fig. 3 h). N-cadherin cell surface expression was quantified by densitometric measurement of cell surface N-cadherin relative to total N-cadherin (Fig. 3 i). We found that co-expression of WT δ-catenin with N-cadherin resulted in a 2.82-fold increase in N-cadherin cell surface expression (Fig. 3 j). The co-expression of G810R polymorphism similarly resulted in a 2.59-fold increase in N-cadherin cell surface expression (Fig. 3 j). In contrast, the R713C mutation, which altered the charge of the N-cadherin JMD binding groove, failed to improve N-cadherin cell surface expression as the surface expression of N-cadherin was not different from the control condition of N-cadherin overexpression in HEK293 cells (Fig. 3 j).

Our protein modeling and biochemical experiments reveal that δ-catenin binds directly to N-cadherin JMD to regulate cadherin cell surface expression. δ-Catenin ARM domain folds into a solenoid structure with a positively charged groove critical for cadherin–catenin complex function. The autism-linked R713C point mutation does not impair N-cad/δ-catenin interaction but is sufficient to impair the function of δ-catenin in enhancing N-cadherin cell surface expression.

## δ-Catenin regulates astrocyte morphogenesis through cadherins

Next, we tested if δ-catenin regulates astrocyte morphogenesis through cadherins using the cortical astrocyte–neuron co-culture assay. Wild-type astrocytes were transfected with shCtnnd2 along with WT, R713C, or G810R before being co-cultured on wild-type neurons for 48 h and assessed for morphological complexity. Congruent with the previous rescue experiment (Fig. 2, d and e), silencing δ-catenin expression severely stunts astrocyte morphology which is rescued by overexpression of full-length δ-catenin (Fig. 4, a and b). Overexpression of either WT or G810R δ-catenin similarly rescued astrocyte complexity. On the contrary, overexpressing R713C could not fully rescue the effect of shCtnnd2, and astrocytes still displayed a simple morphology, as evidenced by the lack of secondary processes and the absence of tertiary processes (Fig. 4, a and b). However, these astrocytes were slightly more complex than Ctnnd2 knockdown astrocytes (P = 0.047). This result strongly suggests that the function of δ-catenin in regulating cadherin surface levels is required for its role in astrocyte morphogenesis, indicating that cadherin function is also required for astrocyte morphogenesis.

To further test this possibility, we investigated whether silencing Ctnnd2 in purified astrocyte cultures is sufficient to decrease the surface expression of endogenous N-cadherin (Fig. 4 c), similar to what was observed in HEK293T cultures (Fig. 3, i and j). Nucleofection of shCtnnd2 resulted in a 52.5 ± 0.03% reduction (P = 0.004) in δ-catenin expression in astrocyte lysates compared with shControl-transfection (Fig. 4 d and Fig. S3 a). Knockdown of Ctnnd2 did not significantly change total N-cadherin expression in astrocytes (P = 0.13, Fig. 4 d and Fig. S3 b). Strikingly, however, there was a significant 62.4 ± 0.1% decrease in surface expression of endogenous N-cadherin in shCtnnd2 compared to shControl astrocytes (P = 0.023, Fig. 4, d and e). Taken together, these data reveal that Ctnnd2 controls surface cadherin expression in astrocytes.

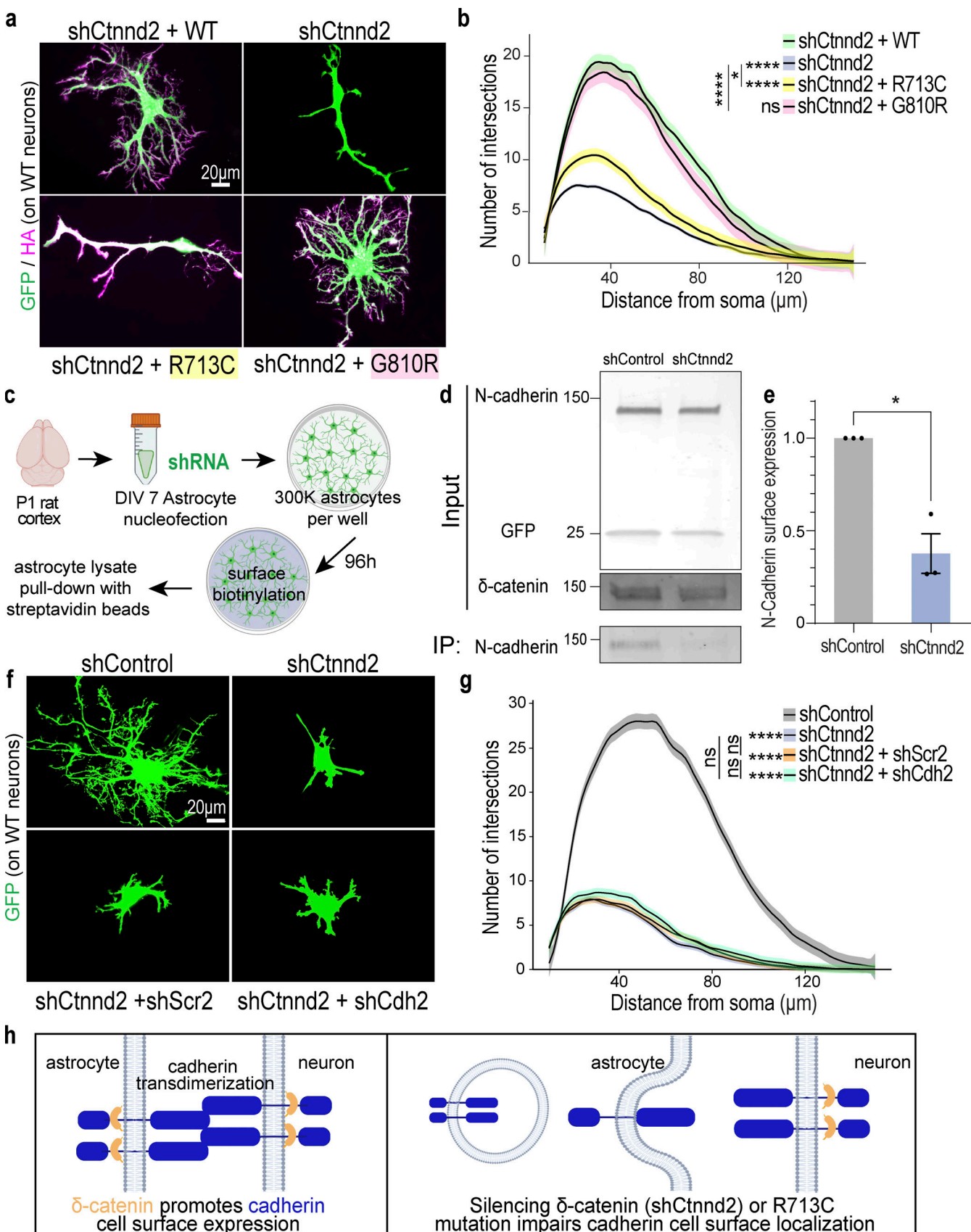

Figure 4. **Astrocyte morphogenesis is regulated by the cadherin–catenin adhesion complex. (a)** Representative images of rat astrocytes co-transfected with shCtnnd2 (green) and human δ-catenin WT, R713C, or G810R (magenta) after 48 h co-culture with wild-type neurons (not labeled). **(b)** Autism-linked

R713C mutation abolishes the ability of δ-catenin to control neuron-contact-dependent astrocyte morphogenesis. Complexity of shCtnnd2 (P < 2.2 × 10⁻¹⁶) and shCtnnd2 + R713C astrocytes (P < 2.2 × 10⁻¹⁶) but not shCtnnd2 + G810R (P = 0.06) astrocytes were significantly reduced when compared with shCtnnd2 + WT astrocytes. n = 92–115 astrocytes per condition from three independent experiments. Linear mixed model with Tukey HSD. ns, not significant. **(c)** Schematic of surface biotinylation experiment in purified astrocyte cultures to determine if knockdown of astrocytic Ctnnd2 alters N-cadherin cell surface expression. Astrocytes were isolated from P1 rat cortices to obtain purified astrocyte cultures. DIV 7 astrocytes are nucleoporated with pLKO.1 plasmid expressing hU6-shRNA and CAG-EGFP and seeded at a density of 300,000 per well of a 6-well plate. Nucleoporated astrocytes were cultured in AGM for 96 h to reach confluency before surface biotinylation and pull-down with streptavidin beads. **(d)** Immunoblot of N-cadherin cell surface expression in astrocyte lysates nucleoporated with shControl or shCtnnd2. All protein molecular weights are expressed in kiloDaltons (kD). **(e)** Silencing astrocytic Ctnnd2 significantly reduces N-cadherin expression at the cell surface (P = 0.023). n = 3 independent replicates. Unpaired t test with Welch's correction. **(f)** Representative images of rat astrocytes co-transfected with shCtnnd2 (green) and shCdh2 or shScr2 (scramble of shCdh2) after 48 h co-culture with wild-type neurons (not labeled). **(g)** Cdh2 and Ctnnd2 are epistatic to one another. Transfection of shCdh2 (P = 0.09) or shScr2 (P = 0.70) in Ctnnd2 knockdown astrocytes did not further reduce astrocyte morphology. No statistical difference in astrocyte complexity was observed in astrocytes co-transfected with shCtnnd2 and shCdh2 or shCtnnd2 and shScr2 (P = 0.57). n = 92–98 astrocytes per condition from three independent experiments, linear mixed model with Tukey HSD. ns, not significant. **(h)** Schematic of working model based on in vitro findings. All data are presented as mean ± SEM. Scale bars: 20 μm. * P < 0.05; **** P < 0.0001. Source data are available for this figure: SourceData F4.

Next, we tested if silencing cadherin expression in astrocytes would alter astrocyte morphology. We focused on Type I and Type II classical cadherins because they possess the JMD necessary for binding to δ-catenin (Oda and Takeichi, 2011). Of the 18 different classical cadherins (Polanco et al., 2021), we identified four cadherins (Cdh2/N-Cadherin, Cdh10, Cdh11, and Cdh20) to be expressed abundantly by cortical astrocytes in both murine and human brains (Zhang et al., 2014, 2016; Farhy-Tselnicker and Allen, 2018). We developed shRNAs targeting each of the four cadherins and validated their knockdown efficiency (Fig. S2 c). Then we utilized them in the cortical astrocyte–neuron co-culture assay to determine how silencing cadherins in astrocytes would impact neuron-contact-dependent astrocyte morphogenesis.

Individual knockdown of astrocytic Cdh2, Cdh10, Chd11, or Cdh20 all stunted astrocyte morphological maturation, albeit to varying degrees (Fig. S3 d and Fig. 3 e). These astrocytes tended to have fewer or shorter primary and secondary branches, unlike shControl astrocytes which possessed primary, secondary, and tertiary branches that extended outwards (Fig. S4 d and Fig. S5). Quantification of astrocyte complexity by Sholl analysis also supported this observation. shCdh2, shCdh10, shCdh11, and shCdh20 astrocytes were significantly less complex than shControl (Fig. S4 e). In addition, while cadherin knockdown astrocytes had a higher peak number of intersections compared to shCtnnd2 astrocytes, the complexity of shCdh2, shCdh10, shCdh11, and shCdh20 astrocytes were statistically similar to shCtnnd2 astrocytes (P = 0.91, P = 0.94, P = 0.99, P = 0.06, respectively, Tukey Honestly Significant Difference [HSD]). These results show that astrocytic cadherins are required for neuronal-contact-dependent astrocyte morphogenesis in vitro.

We also performed co-knockdown of Ctnnd2 and Cdh2 (NCad) in astrocytes to test whether δ-catenin works in the same pathway as cadherins in astrocytes to regulate neuron-contact dependent astrocyte morphogenesis in vitro. Indeed, we found no statistical difference in astrocyte complexity between Ctnnd2 knockdown astrocytes transfected with shCdh2 and its scramble construct, shScr2 (P = 0.57, Fig. 4 g). Furthermore, we also noted that Ctnnd2 and Cdh2 double knockdown astrocytes only had a few short primary processes, characteristic of shCtnnd2 astrocytes (Fig. 4 f). Comparison of both conditions by Sholl analysis confirmed that knockdown of both Ctnnd2 and Cdh2 did not enhance the morphogenesis defect (P = 0.70,

Fig. 4 g) and is indicative that Ctnnd2 and Cdh2 work in the same molecular pathway to regulate astrocyte morphogenesis.

Taken together, our in vitro findings suggest the following model (Fig. 4 e): Astrocytes and neurons use cadherins to establish transcellular adhesions, which are required for astrocytes' ability to gain their complex morphology in response to neuronal contact. δ-Catenin is required both in astrocytes and in neurons for maintaining cadherin cell-surface localization. Therefore, when δ-catenin is silenced in astrocytes or the autism-linked R713C mutation is present, cadherin-based astrocyte–neuron interactions are impaired, leading to deficient astrocyte morphogenesis.

### Loss of astrocytic δ-catenin is sufficient to reduce astrocyte complexity in vivo

Our in vitro findings pointed out a critical role for δ-catenin in astrocyte morphogenesis during development. Next, we determined if δ-catenin regulates astrocyte morphogenesis in vivo by comparing the morphology of shControl and shCtnnd2 transfected astrocytes in the primary visual V1 cortex across development (P7, P14, and P21). For these experiments, the shRNAs were cloned into a PiggyBac transposon system expressing mCherry-CAAX and introduced into radial glia cells at P0 by postnatal astrocyte labeling by electroporation (PALE), resulting in a sparse knockdown and labeling of cortical astrocytes (Fig. 5 a). Whole astrocytes were imaged by confocal microscopy and morphologies were reconstructed using Imaris (Bitplane). Astrocyte morphogenesis in vivo was quantified in two ways: whole astrocyte morphological complexity measured by 3D Sholl analysis and astrocyte territory volume analyzed by convex hull analysis (Fig. 5 b). In general, we found that silencing Ctnnd2 through shRNA in vivo utilizing the PALE method decreased δ-catenin expression by 41% (P = 0.031, Fig. S4 a and Fig. 4 b).

First, we noted that silencing δ-catenin expression did not affect astrogenesis as we found mCherry-positive astrocytes in mice electroporated with shCtnnd2. This was a concern because δ-catenin is expressed in neural progenitors (Ho et al., 2000). We could also detect labeled radial glia processes spanning the cortex at P4 and P7. The radial glia processes and mCherry-positive astrocytes across all cortical layers suggest our manipulation did not perturb astrocyte migration. Cortical shCtnnd2 astrocytes were significantly less complex (Fig. 5 c) and smaller

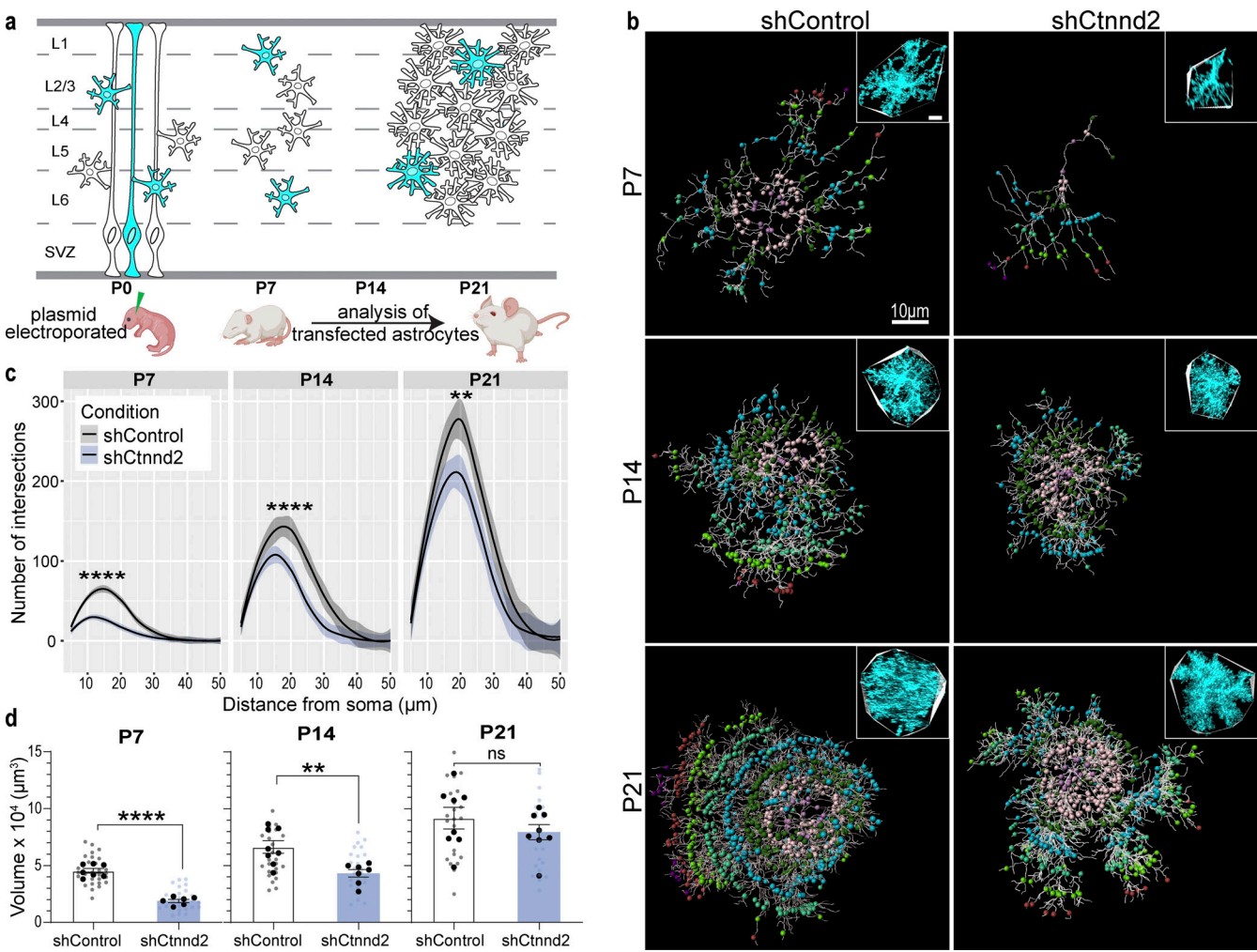

Figure 5. **Loss of astrocytic δ-catenin is sufficient to reduce astrocyte complexity in vivo. (a)** Schematic of PALE. Plasmids were injected into the lateral ventricle of CD1 mice and electroporated into radial glial stem cells at late P0 resulting in a sparse knockdown and labeling of cortical astrocytes. Brains were collected at P7, P14, and P21 to analyze astrocyte morphological complexity and territory. SVZ, subventricular zone. **(b)** Representative images of in vivo astrocytes transfected shControl or shCtnnd2-1. Whole astrocytes were reconstructed using the Imaris filament tracing tool. Inset shows a confocal image of the astrocyte with a convex hull denoting astrocyte territory volume. **(c)** Silencing astrocytic δ-catenin resulted in a significant decrease in astrocyte complexity at P7 (P = 1.38 × 10$^{-14}$), P14 (P = 1.68 × 10$^{-5}$), and P21 (P = 6.90 × 10$^{-3}$). Quantification of in vivo astrocyte complexity with 3D Sholl analysis. n = 23–28 astrocytes from six to eight mice. ANOVA, linear mixed model with Tukey HSD. **(d)** Silencing astrocytic δ-catenin resulted in a significant decrease in astrocyte territory volume at P7 (P < 0.0001), P14 (P = 0.0027), but not P21 (P = 0.18). Quantification of in vivo astrocyte territory volumes by convex hull analysis. Only astrocytes from V1 cortex were imaged and analyzed. The average astrocyte territory volume of individual mice is plotted in black. n = 23–28 astrocytes from six to eight mice, nested t test for each time point. ns, not significant. All data is presented as mean ± SEM. Scale bars: 10 μm. ** P < 0.01; **** P < 0.0001.

(Fig. 5 d) than shControl astrocytes at P7 and P14. Both shCtnnd2 and shControl astrocytes matured through brain development, as evidenced by the stepwise increase in astrocyte complexity and territory volume from P7 to P21 (Fig. 5, c and d). However, shCtnnd2 astrocytes could catch up with shControl astrocytes in terms of territory volume at P21 (Fig. 5 d), but they were still significantly less complex (Fig. 5 c).

We then further characterized P14 shControl and shCtnnd2 astrocytes because δ-catenin expression peaks at P14 (Fig. 1 b), and we observed the strongest phenotype in vivo at this time point. Silencing Ctnnd2 expression significantly reduced astrocyte complexity of both upper-layer (P = 0.015) and lower-layer (P = 4.6 × 10$^{-4}$) astrocytes (Fig. S4 c). However, astrocyte

territory volume only decreased significantly in upper-layer shCtnnd2 astrocytes (P = 0.0015, Fig. S4 d). Because reduced astrocyte morphological complexity has been linked to changes in synapse number and function (Stogsdill et al., 2017; Takano et al., 2020; Baldwin et al., 2021), we quantified the density of excitatory synapses based on colocalization of presynaptic marker Basson and postsynaptic marker PSD95 within a region of shControl and shCtnnd2 astrocytes (Fig. S4 e). We did not observe any difference in the density of excitatory synapses (P = 0.72, Fig. S4 f and Fig. 4 g). Therefore, we concluded that δ-catenin regulates astrocyte complexity in vivo, where a loss of astrocytic δ-catenin results in decreased complexity throughout development and across all cortical layers.

## Neuronal δ-catenin is required for astrocyte complexity

Our in vitro findings suggested that the cadherin–catenin adhesion complexes between astrocytes and neurons mediate astrocyte morphogenesis (Fig. 4 e). In this model, we expect δ-catenin to regulate cadherin surface localization both in astrocytes and neurons. Therefore, we hypothesize that δ-catenin function is required both in astrocytes and in neurons to mediate astrocyte morphology. To test this hypothesis, next, we silenced δ-catenin expression only in the upper-layer neurons of the developing cortex and examined how this neuronal manipulation affects wild-type astrocyte morphogenesis. If astrocyte–neuron cadherin-based interactions regulated astrocyte morphogenesis, we expect that loss of upper-layer neuronal cadherins due to silencing of δ-catenin would reduce the complexity of only the upper-layer but not the lower-layer astrocytes (Fig. 6 a).

We electroporated shCtnnd2 or shControl into radial glial cells in utero at E15.5 to silence δ-catenin expression, specifically in upper-layer neurons. After birth (late P0), we performed PALE to introduce mCherry-CAAX into a sparse population of wild-type astrocytes in the primary visual cortex (Fig. 6 a). Mice were sacrificed at P21 after the conclusion of astrocyte morphological maturation. The brain sections were costained with Ctip2, a lower-layer neuronal marker, to help distinguish between upper- and lower-layer astrocytes (Fig. 6 b).

In line with our model, we found a severe reduction in upper-layer astrocyte complexity (Fig. 6, c and d) and a 0.39-fold decrease in astrocyte territory volume (Fig. 6, c and e) when the δ-catenin expression was knocked down in upper-layer neurons. Neither a change in astrocyte morphology nor territory volume was observed in the lower layer astrocytes, which were surrounded by unmanipulated neurons (Fig. 6, d and e). This result reveals that neuronal δ-catenin regulates astrocyte morphogenesis locally in a non-cell autonomous manner.

## N-cadherin/Cdh2 is specifically required for lower-layer astrocyte morphogenesis

Silencing δ-catenin in upper-layer neurons only affected the morphology of upper-layer astrocytes, strongly supporting the model that astrocyte morphogenesis is mediated through transcellular astrocyte–neuron cadherin interactions. This result also suggests that different cadherins may mediate interactions of astrocytes with distinct subsets of neurons. To explore this possibility, we reanalyzed the single-cell transcriptomic datasets of the P14 cortical neurons (Stogsdill et al., 2022) and astrocytes (Farhy-Tselnicker et al., 2021) to map out Type I and Type II classical cadherin expression in neurons and astrocytes across the cortical layers. Type I cadherins form high-affinity homophilic trans-dimers, whereas Type II cadherins form both homophilic and heterophilic trans-dimers within the members of the same specificity group (Brasch et al., 2018; Polanco et al., 2021). We excluded lowly expressed cadherins from our analysis (<10 fragments per kilobase of transcript per million mapped reads, FPKM) and charted cadherin expression in astrocytes and neurons according to cortical layers (L2/3 and L4 = upper layer, L5 and L6 = lower layer). This resulted in a putative cortical layer-specific astrocyte–neuron cadherin binding code (Fig. S3). We noticed that Cdh2/N-cadherin expression is high in all

cortical astrocytes but restricted to lower-layer cortical neurons. If astrocyte-neuron N-cadherin trans-dimerization is required for astrocyte morphogenesis, we hypothesized that N-cadherin expression in astrocytes is only important for lower-layer cortical astrocyte complexity because only the lower-layer neurons express this cadherin (Fig. 7 a).

To test this possibility, we silenced N-cadherin expression in a sparse population of astrocytes by introducing shNCad (same as shCdh2) by PALE at late P0. Both upper- and lower-layer astrocytes were labeled in this experimental paradigm, and we analyzed astrocyte complexity and astrocyte territory volume in shControl and shNCad astrocytes at P14 and P21 (Fig. 7 b). At P14, N-cadherin knockdown lower-layer astrocytes were 30% smaller than control lower-layer astrocytes, but no change in astrocyte territory was observed between upper-layer astrocytes (Fig. 7 c). Like shCtnnd2 astrocytes, shCdh2 astrocytes were able to grow across development. By P21, there was no statistical difference in average astrocyte territory volume between shNCad and shControl astrocytes when comparing upper-layer or lower-layer astrocytes (Fig. 7 d).

In contrast, there was a highly significant reduction in lower-layer astrocyte complexity throughout development when N-cadherin expression was silenced in astrocytes (Fig. 7 e). Upper-layer astrocyte complexity remained unchanged (Fig. 7 e), highlighting the necessity of N-cadherin exclusively for lower-layer astrocyte morphogenesis. This effect remained consistent throughout development at P7 and P21. Taken together, these data reveal that transcellular astrocyte–neuron interactions mediated by cadherin trans-dimerization control astrocyte morphogenesis in a layer-specific manner.

## Discussion

Astrocytes play many essential roles in brain circuit formation and function (Chung et al., 2015; Farhy-Tselnicker and Allen, 2018; Tan et al., 2021), and this is dependent on their complex morphology, which enables astrocytes to interact with synapses and other elements of the brain parenchyma. Still, we are only starting to discover developmental mechanisms that explain how astrocytes gain their complex, arbor-like morphologies. Recent work tracking murine cortical astrocyte development through clonal analysis, image reconstructions, and transcriptional profiling revealed cortical-layer specific morphological heterogeneity and suggested that astrocyte morphological maturation is independent of progenitor identity but is instead linked to the neuronal organization and local environmental cues (Lanjakornsiripan et al., 2018; Clavreul et al., 2019; Bayraktar et al., 2020; Torres-Ceja and Olsen, 2022). Cell adhesion molecules guide astrocyte morphogenesis, and the disruption of astrocyte–neuron contact restricts the astrocyte process growth (Stogsdill et al., 2017; Takano et al., 2020; Baldwin et al., 2021). However, none of these molecules regulate astrocyte complexity in a layer-specific manner, suggesting that there are other essential mechanisms underlying layer-specific astrocyte maturation.

Here, we present a new mechanism that controls astrocyte morphogenesis in a layer-specific manner. We found that

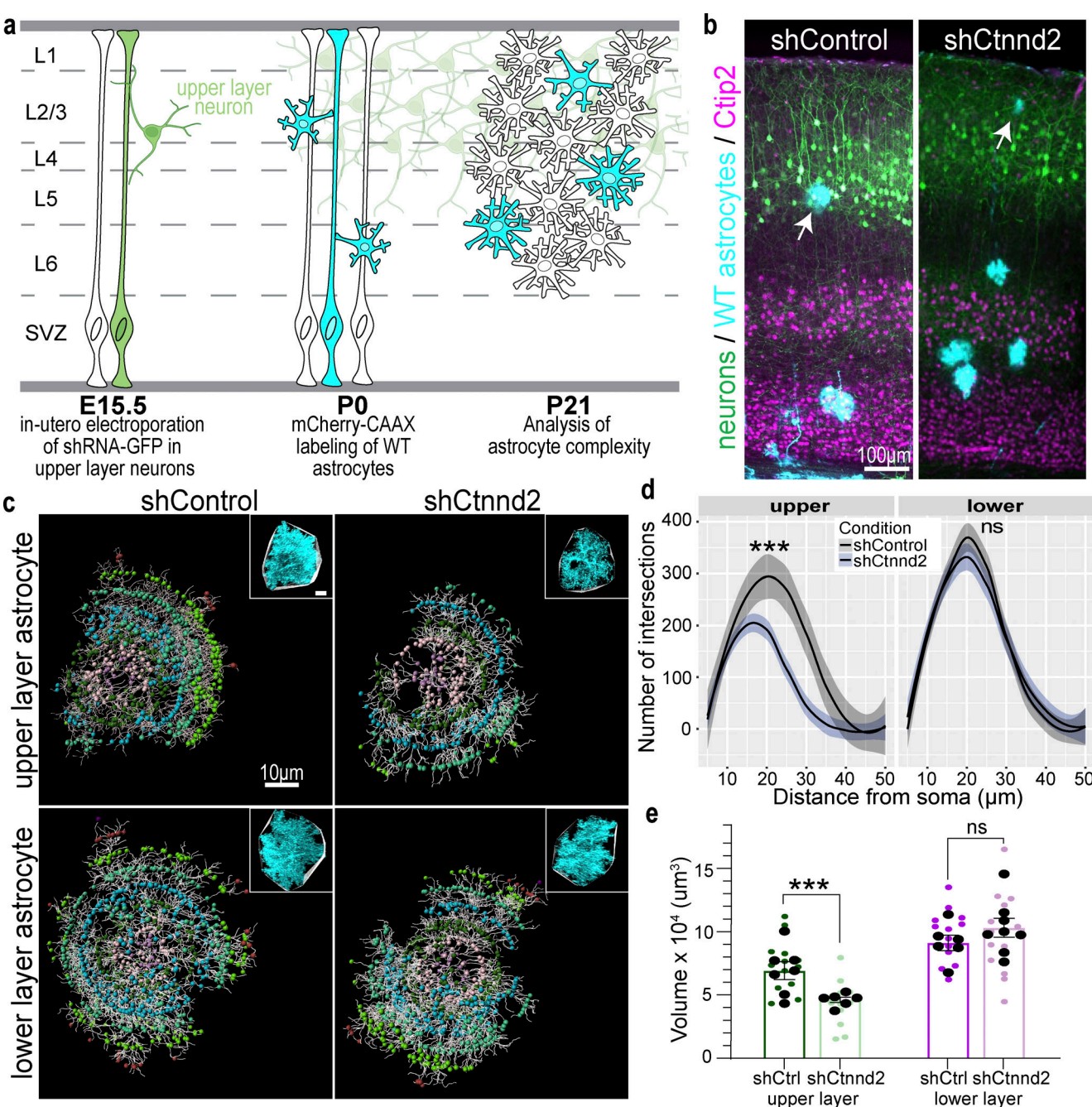

Figure 6. **Neuronal δ-catenin is required for astrocyte complexity. (a)** Schematic of experimental design. *Ctnnd2* was silenced in upper-layer neurons by IUE at E15.5. At P0, wild-type astrocytes were labeled by mCherry-CAAX (cyan) by PALE. Brains were collected at P21 for analysis of astrocyte morphology and territory size. SVZ, subventricular zone. **(b)** Representative images of the primary visual cortex after IUE and PALE. Upper-layer neurons (green) are transfected with shControl or shCtnnd2, lower-layer neurons (magenta) are labeled with Ctip2, and wild-type astrocytes (cyan) are labeled with mCherry-CAAX. **(c)** Representative images of P21 astrocytes after upper-layer neurons were transfected with shControl or shCtnnd2. Whole astrocytes were reconstructed using the Imaris filament tracing tool. Inset shows a confocal image of the astrocyte with a convex hull denoting astrocyte territory volume. **(d)** Silencing δ-catenin expression in upper-layer neurons resulted in a significant decrease in upper-layer astrocyte complexity (P = 5.99 × 10⁻⁴) but not in lower-layer astrocyte complexity (P = 0.96). Quantification of in vivo astrocyte complexity with 3D Sholl analysis. *n* = 13–17 astrocytes from six to nine mice per condition. ANOVA, linear mixed model with Tukey HSD. ns, not significant. **(e)** Silencing δ-catenin expression in upper-layer neurons resulted in a significant decrease in upper-layer astrocyte territory volume (P = 0.0005) but not in lower-layer astrocyte complexity (P = 0.45). Quantification of in vivo astrocyte territory volumes by convex hull analysis. Astrocytes from the upper (green) and lower purple) layers of the V1 cortex were imaged and analyzed. Average astrocyte territory volume of individual mice is plotted in black. *n* = 13–17 astrocytes from six to nine mice. Nested *t* test for each layer. ns, not significant. All data are presented as mean ± SEM. The scale bar in b is 100 µm, while the scale bar in c is 10 µm. *** P < 0.001.

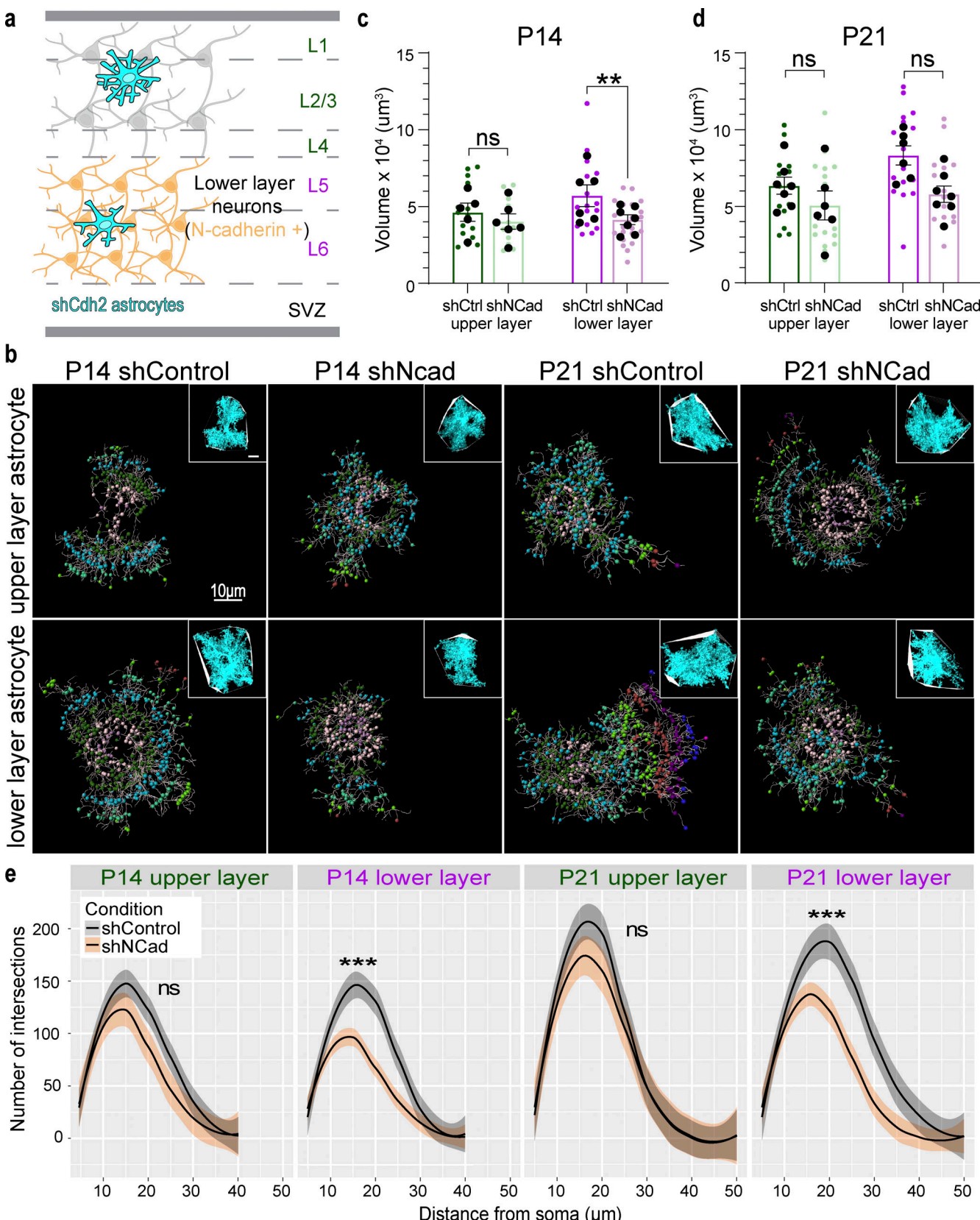

Figure 7. **N-Cadherin/Cdh2 is specifically required for lower-layer astrocyte morphogenesis. (a)** Schematic representation of astrocytic and neuronal N-cadherin expression where only lower-layer neurons and all cortical astrocytes express N-Cadherin (orange). We hypothesize that the loss of astrocytic N-cadherin (cyan) by silencing *Cdh2* expression will more strongly impact lower-layer astrocyte morphogenesis and territory size. SVZ, subventricular zone. **(b)** Representative images of in vivo astrocytes transfected with shControl (scramble sequence of shNcad) or shRNA construct targeting Cdh2 (shNcad) at P14

and P21. Whole astrocytes were reconstructed using the Imaris filament tracing tool. Inset shows confocal image of the astrocyte with a convex hull denoting astrocyte territory volume. **(c)** Silencing N-cadherin expression resulted in a significant decrease in P14 lower-layer astrocyte territory volume (P = 0.0076) but not in upper-layer astrocytes (P = 0.47). Quantification of P14 in vivo astrocyte territory volumes by convex hull analysis. Astrocytes from the upper (green) and lower (purple) layers of the V1 cortex were imaged and analyzed. Average astrocyte territory volume of individual mice is plotted in black. $n$ = 11–21 astrocytes from five to seven animals per condition. Nested $t$ test for comparison within each layer. ns, not significant. **(d)** Neither upper-layer (P = 0.37) nor lower-layer (P = 0.12) astrocyte territory volume was significantly reduced by N-cadherin knockdown. Quantification of P21 in vivo astrocyte territory volumes by convex hull analysis. Astrocytes from the upper (green) and lower (purple) layers of the V1 cortex were imaged and analyzed. Average astrocyte territory volume of individual mice is plotted in black. $n$ = 14–17 astrocytes from five to seven animals per condition. Nested $t$ test for comparison within each layer. ns, not significant. **(e)** Silencing N-cadherin expression results in severe loss of complexity in lower-layer astrocytes at P14 (P = $8.35 \times 10^{-4}$) and P21 (P = $3.18 \times 10^{-4}$). In contrast, upper-layer astrocyte morphology remains unperturbed at P14 (P = 0.094) and P21 (P = 0.48). Quantification of in vivo astrocyte complexity of P14 and P21 astrocytes with 3D Sholl analysis. $n$ = 11–21 astrocytes from five to seven animals per condition for P14 and $n$ = 14–17 astrocytes from five to seven animals per condition for P21. ANOVA, linear mixed model with Tukey HSD. ns, not significant. All data are presented as mean ± SEM. The scale bar is 10 µm. ** P < 0.01; *** P < 0.001.

δ-catenin is expressed in astrocytes and that the cadherin–catenin adhesion complex is a critical regulator of astrocyte morphological maturation. Silencing δ-catenin in either astrocytes or neurons diminishes cadherin cell surface expression, preventing cadherin trans interactions between astrocytes and neurons. In addition, while neuronal δ-catenin is essential for astrocytic morphogenesis, astrocytic δ-catenin may be dispensable for neuronal morphogenesis and excitatory synapse formation. We also found that cortical astrocytes and neurons have layer-specific cadherin expression during development, which could form the basis of a chemo-affinity code between astrocyte–neuron contacts and, therefore, regulate layer-specific astrocyte morphogenesis.

Our findings align with the hypothesis that layer-specific neuronal cues give rise to layer-specific astrocyte heterogeneity (Lanjakornsiripan et al., 2018; Bayraktar et al., 2020) and suggest that a neuron–glia cadherin code serves an essential molecular mechanism underlying astrocyte specialization. In this model, the neurons, which are born first, establish the cadherin code within cortical layers, and astrocytes that arrive later utilize this code to direct their morphogenesis. However, it is critical to acknowledge the existence of astrocyte and neuronal diversity within a single cortical layer, the most notable of which are the different neuron subpopulations named for the type of projections they make (Stogsdill et al., 2022). Furthermore, various combinations of neuronal cadherins are already known to direct the assembly of specific circuits (Duan et al., 2014, 2018; Frei et al., 2021; Vagnozzi et al., 2022). An intriguing hypothesis remains untested: Does the cadherin-based adhesion code between neurons and astrocytes confer circuit specificity? In this scenario, cadherin trans interactions between astrocytes and subsets of axons and dendritic spines promote the formation and function of particular circuits.

Identifying the cadherin–catenin adhesion complex as a crucial regulator of astrocyte morphogenesis also mechanistically links cell–cell contact with actin cytoskeleton remodeling, explaining why neuron contact is necessary for astrocyte complexity. The canonical cadherin–catenin complex comprises other effector proteins downstream from cadherins and δ-catenin. α-Catenin is another crucial component of the complex and binds to the cytoplasmic tail of cadherins. α-Catenin directly binds to actin and regulates actin assembly and dynamics required for actin cytoskeletal changes (Shapiro and Weis, 2009). It is also well-established in hippocampal neuron

cultures that δ-catenin binds to small Rho-GTPases to modulate crosstalk between Rac and Rho signaling pathways that regulate cadherin function and actin dynamics necessary for dendrite morphology (Kim et al., 2008; Baumert et al., 2020; Donta et al., 2022). A similar molecular mechanism may also play a critical role in astrocyte morphogenesis.

Another aspect of the cadherin–catenin adhesion complex is its ability to signal to the nucleus and affect transcription through β-catenin. In the canonical adherens junction, cytosolic β-catenin is recruited to the plasma membrane and its binding to E-cadherin is required for cell–cell adhesion (Yap et al., 1997). β-Catenin is also an essential component of the Wnt signaling pathway: When Wnt ligands bind to the Frizzled receptor, the APC complex that degrades cytosolic β-catenin is inhibited, resulting in the accumulation of cytosolic β-catenin, nuclear translocation, and binding to transcription factors to promote gene expression (Logan and Nusse, 2004). β-Catenin serves as an integrating signal of cell–cell adhesion with gene expression, especially since cadherin-bound β-catenin was demonstrated to be internalized together with E-cadherin in the perinuclear endocytic recycling compartment and translocated into the nucleus upon Wnt pathway activation in epithelial cells (Kam and Quaranta, 2009). Furthermore, it has been postulated that β-catenin is primed for Wnt signaling after binding with and dissociation from cadherins (Howard et al., 2011). This crosstalk between cell adhesion and Wnt signaling allows for developing cells to migrate and acquire new cell fates during embryogenesis (Nelson and Nusse, 2004), and δ-catenin overexpression has been demonstrated to drive β-catenin signaling in human prostate cancer (Kim et al., 2012; Nopparat et al., 2015).

It is plausible that this crosstalk is similarly critical during astrocyte maturation. As astrocytes gain complex, arbor-like morphologies from P7–P21, they also switch from proliferative to mature transcriptional states, upregulating genes necessary for adult brain homeostasis, neuronal trophic support, and modulation of neuronal activity (Lattke et al., 2021). Despite the importance of astrocyte morphological and transcriptional maturation, it is not known whether these two processes are mechanistically linked and what the underlying mechanism might be. We propose that cadherin trans-dimerization between astrocytes and neurons not only regulates layer-specific astrocyte morphogenesis but also specializes astrocyte transcription through β-catenin downstream of the cadherin–catenin adhesion complex. Our in vivo experimental paradigm of silencing

δ-catenin in a sparse population of astrocytes limited us from collecting sufficient cells needed for bulk RNA transcriptomic analysis, but if our hypothesis is correct, we expect morphologically simple shCtnnd2 astrocytes to be transcriptionally immature at P21. Taken together, δ-catenin regulates multiple facets of astrocyte development: it is required for cadherin-based astrocyte heterogeneity and is necessary for layer-specific astrocyte morphogenesis. Future studies investigating the transcriptional impacts of cadherin–catenin-based interactions between astrocytes and neurons are poised to link astrocyte morphogenesis to transcriptional maturation mechanistically.

Finally, this study has important implications for understanding the pathophysiology of neurological diseases, particularly neurodevelopmental diseases that δ-catenin is associated with, such as autism spectrum disorder and intellectual disability (Medina et al., 2000; Sardina et al., 2014; Belcaro et al., 2015; Hofmeister et al., 2015; Miller et al., 2020). Mutations in δ-catenin are most strongly correlated to cognitive deficits, which have been attributed solely to synaptic deficits (Israely et al., 2004; Abu-Elneel et al., 2008; Matter et al., 2009; Arikkath et al., 2009; Yuan et al., 2015). Our findings reveal that δ-catenin is significant in glial development and necessary for astrocyte–neuron bilateral signaling. Since astrocytes directly regulate synapse formation and function (Chung et al., 2015; Farhy-Tselnicker and Allen, 2018; Tan et al., 2021), astrocyte dysfunction may underlie some of the synaptic deficits previously observed. In conclusion, we show a novel function of δ-catenin in astrocytes. δ-Catenin is required for cadherin cell-surface expression in astrocytes and neurons and regulates layer-specific neuron-contact-dependent astrocyte morphological maturation.

## Materials and methods

### Animal studies
All mice and rats were used in accordance with the Institutional Animal Care and Use Committee and with oversight by the Duke Division of Laboratory Animal Resources under Institutional Animal Care and Use Committee Protocol Numbers A147-17-06 and A117-20-05. All animals were housed under 12-h light/dark cycles. Aldh1L1-EGFP (RRID: MMRRC_011015-UCD) mice were obtained through MMRRC and backcrossed on a C57BL/6J background. Wild-type C57BL/6J (RRID: IMSR_JAX:000664) mice were obtained from Jackson Laboratories. Timed-pregnant wild-type CD1 mice (RRID: IMSR_CRL:022) used for in-utero electroporation (IUE) and PALE while CD (Sprague-Dawley) IGS rats (SD-001) used for primary culture preps were purchased from Charles River Laboratories. Mice and rats of both sexes were included in all experiments. Sex was not an influence on any of the experimental outcomes.

### Plasmids
#### shRNA plasmids
pLKO.1 puro plasmids containing shRNA that target the following genes were purchased from RNAi Consortium (TRC) via Dharmacon: Ctnnd2 (5′-ATTGGAGGTGTTTAAGCCTGG-3′ and

5′-ATGAAGAGAGTAAACAAAGCG-3′), Cdh2 (5′-ATTGAGTAT TCTAACTACAGC-3′), Cdh10 (5′-ATGAGAACCGAAGACTATATA-3′), Cdh11 (5′-ATTGTATGGGTTTAAGCTGGC-3′), and Cdh20 (5′-AGGAGCAGGCATCGTGTTTAC-3′).

The scrambled sequence of each shRNA was generated in the lab and cloned into the pLKO.1 TRC cloning vector according to Addgene protocols: Ctnnd2 (5′-GCCTTGGGAGTTAAGTGATGT-3′ and 5′-GAGGACAAATAAGGGAATCAA-3′), Cdh2 (5′-GAACTC TTAATCGCAATATTG-3′), Cdh10 (5′-GGACATTAACGAATAATA CAG-3′), Cdh11 (5′-AGCCGTGATTGAGTATTGTTG-3′), and Cdh20 (5′-GTCGTGTACACGTCAAGGAGT-3′). All shRNA sequences were verified by Basic Local Alignment Search Tool alignment to be specific to mouse and rat isoforms of target gene and all scramble sequences were verified to have no known gene targets. pLKO.1 shRNA plasmids that express EGFP were synthesized from the pLKO.1 puro by replacing the puromycin resistance gene with CAG-EGFP. pPB-shRNA-mCherryCAAX plasmids were synthesized by DNA amplification of hU6 promoter and shRNA from pLKO.1-shRNA-GFP using Phusion High-Fidelity DNA Polymerase (#M0530S; NEB) using primers that also introduced SpeI restriction site (forward primer: 5′-GGACTAGTCAGGCCCGAAGGAATAGAAG-3′; reverse primer: 5′-GGACTAGTGCCAAAGTGGATCTCTGCTG-3′). After PCR purification and digestion with SpeI, the resulting hU6 and shRNA DNA fragment were ligated into pPB-mCherryCAAX.

#### δ-Catenin plasmids
Synthetic construct human CTNND2 cDNA encoding for the complete protein (Clone ID 40080647; Dharmacon) was cloned into gateway destination vector pDest-pCS2+. Two gene blocks were synthesized to insert the NheI restriction site upstream of the cDNA and to insert an HA tag (5′-TACCCATACGATGTTCCA GATTACGCT-3′) immediately before the stop codon. The resulting construct was digested with restriction enzymes NheI and BmtI, purified, and ligated into pZac2.1-gfaABC1D vector that had an astrocyte-specific promoter. This construct is termed pZac2.1-gfaABC1D-hCTNND2-HA and used as the full-length rescue construct in rescue experiments. Mutations R713C (rs768575356, G > A) and G810R (rs61754599, C > G) were cloned into pZac2.1-gfaABC1D-hCTNND2-HA by generating gene blocks containing the respective mutation between SalI and AatII restriction cut sites. All δ-catenin plasmids were sequenced with primers at 500-bp intervals and confirmed to not have other point mutations. HA-tagged, full-length rescue construct, as well as constructs containing mutations R713C and G810R were cloned into pCDNA3.1 vector for biochemical assays using In-fusion cloning (forward primer: 5′-ACCCAAGCTGGCTAGGGCCCACCATGTTTG CGA-3′; reverse primer: 5′-AAACGGGCCCTCTAGTTAAGCGTA ATCTGGAACATCGT-3′). All pCDNA3.1 δ-catenin plasmids were sequenced by whole plasmid sequencing before use.

#### N-cadherin plasmids
The N-cadherin-EGFP plasmid was a gift from Valeri Vasioukhin (plasmid #18870; Addgene). The conserved JMD sequence (KRRDKERQAKQLLIDPEDDVRDNILKYDEEGGGEEDQDYDLSQL QQPDTVEPD) was deleted using Infusion cloning (forward primer: 5′-TATGGATGGCCATCAAGCCCGTGGGA-3′; reverse

primer: 5′-TGATGGCCATCCATACCACAAACATGAGAACA-3′) and verified by whole plasmid sequencing to obtain N-cadherinΔJMD-GFP.

## Cell culture and transfection
### Cortical neuron isolation and culture
Rat cortical neuron cultures were prepared as described: Briefly, P1 rat cortices of both sexes (Sprague-Dawley; Charles River Laboratories) were micro-dissected and digested in papain (~7.5 U/ml, #LK003178; Worthington Biochemical) at 33°C for 45 min. Digested tissue chunks were triturated in low and high ovomucoid (#LS003086; Worthington Biochemical) solutions to obtain single-cell suspensions that were pelleted and resuspended in panning buffer (Dulbecco's phosphate-buffered saline [DPBS; #14287-080; GIBCO] supplemented with insulin [#10977015; Invitrogen] and bovine serum albumin [#A4161; Sigma-Aldrich]) and passed through a 20-μM mesh filter (#03-20/14; Elko). Filtered cells were plated in the following succession: twice on negative panning dishes coated with Bandeiraea Simplicifolia Lectin 1 (#L-1100-5; Vector Laboratories) followed by a panning dish coated with goat anti-mouse IgG+IgM (H+L; #115-005-044; Jackson ImmunoResearch) antibodies and then a panning dish coated with goat anti-rat IgG+IgM (H+L; #112-005-044; Jackson ImmunoResearch) antibodies. Cells remaining from negative panning were incubated on positive panning dishes coated with mouse anti-rat L1 antibodies (prepared in-house from Developmental Studies Hybridoma Bank #ASCS4) for 45 min to bind cortical neurons. Adherent cells were collected, pelleted (11 min at 200 g), and resuspended in serum-free neuron growth media (NGM; Neurobasal, B27 supplement, 2 mM L-Glutamine, 100 U/ml Pen/Strep, 1 mM sodium pyruvate, 4.2 mg/ml Forskolin, 50 ng/ml brain-derived neurotrophic factor, and 10 ng/ml ciliary neurotrophic factor).

For astrocyte–neuron co-culture assays, 100,000 neurons were plated onto 12-mm glass coverslips coated with 10 μg/ml poly-D-lysine (#P1024; Sigma-Aldrich) and 2 μg/ml Laminin (#3400-010-02; R&D Systems) in neuron–astrocyte co-culture assays. Otherwise, 550,000 neurons/well were plated onto 6-well plates similarly coated with 10 μg/ml poly-D-lysine and 2 μg/ml Laminin. All purified neurons were cultured at 37°C in 10% CO$_2$. On DIV 2, half of the media was replaced with NGM Plus (Neurobasal Plus, B27 Plus, 100 U/ml Pen/Strep, 1 mM sodium pyruvate, 4.2 mg/ml Forskolin, 50 ng/ml brain-derived neurotrophic factor, and 10 ng/ml ciliary neurotrophic factor). AraC (10 mM, #C1768; Sigma-Aldrich) was added to inhibit the growth of proliferating contaminating cells. On DIV 3 (16 h after AraC was added), all the media was replaced with NGM Plus. Neurons were fed every 3–4 d by replacing half of the media with NGM Plus. Around 10% astroglial contamination was detected in these cultures after DIV 9.

For neuron morphology assays, 5,000 neurons were plated onto coverslips previously seeded with DIV 7 nucleoporated astrocytes.

### Cortical astrocyte isolation and culture
Astrocytes were purified from neonatal rat cortices. Briefly, P1 rat cortices of both sexes (Sprague-Dawley, Charles River Laboratories) were harvested and treated similarly as described above to obtain single-cell solutions, which were resuspended in astrocyte growth media (AGM: DMEM [#11960; GIBCO], 10% FBS [#10437028; GIBCO], 10 mM hydrocortisone, 100 U/ml Pen/Strep, 2 mM L-glutamine, 5 mg/ml insulin, 1 mM Na pyruvate, and 5 mg/ml N-Acetyl-L-cysteine). 15–20 million cells were incubated in 75 mm² flasks (non-ventilated cap) coated with poly-D-lysine at 37°C in 10% CO$_2$. On DIV 3, non-astrocyte cells were removed by forceful, manual shaking of the closed flasks for 10–15 s until only an adherent monolayer of astrocytes remained. AraC was added to the AGM from DIV 5 through DIV 7 to eliminate any contamination from fibroblasts. On DIV 7, astrocytes were trypsinized (0.05% Trypsin-EDTA, #25300054; GIBCO) and plated into 6-well dishes at a density of 400,000 astrocytes/well.

In the neuron–astrocyte co-culture and astrocyte monoculture assays, passaged astrocytes are transfected with shRNA and/or expression plasmids using Lipofectamine LTX Reagent with PLUS Reagent (#15338100; Invitrogen) per manufacturer's protocol at DIV 9. Briefly, 2 μg total DNA was diluted in Opti-MEM (#11058021; GIBCO) containing Plus Reagent, mixed with Opti-MEM containing LTX (1:2 DNA to LTX ratio), and incubated at room temperature for 30 min. The transfection solution was then added to astrocyte cultures and incubated at 37°C for 2.5 h in 10% CO$_2$. On DIV 11, astrocytes were trypsinized, resuspended in NGM Plus, plated at a density of 20,000 cells/well onto DIV 11 wild-type neurons or astrocytes, and co-cultured for 48 h. 48 h was chosen as a standard co-culture time because characterization by Stogsdill et al. (2017) (extended data Fig. 2) showed that astrocyte complexity in co-culture conditions does not vary much between 24- to 120-h timepoint and tripartite synapses were detected at 48 h.

### Astrocyte nucleofection
Astrocytes were nucleofected at DIV 7 using the Basic Nucleofector Kit for Primary Mammalian Glial Cells (#VPI-1006; Lonza) according to the manufacturer's protocol. Briefly, astrocytes were trypsinized and cell density was determined. For each condition, the appropriate volume for 2 million cells was added to 15-ml tubes, centrifuged at 11 min at 200 g, and the resulting supernatant was removed completely. The cell pellet was resuspended in 100 μl of room temperature Basic Nucleofector Solution for Mammalian Glial cells and 5 μg of shRNA plasmid. The resulting cell/plasmid suspension was immediately transferred into a cuvette and nucleofected using the T-20 program for Nucleofector 2b Device. After nucleofection, 500 μl of pre-equilibrated NGM Plus was added to the cell/plasmid suspension. Cell density of live cells after nucleofection was determined and nucleofected astrocytes were plated at a density of 80,000/coverslip (for neuron morphology) or 300,000/well of a 6-well plate (for astrocyte surface biotinylation assay). For astrocyte mosaic preps, nucleofected astrocytes were first cultured in AGM for 24 h before being switched to NGM Plus for another 48 h.

### HEK293T cell culture
HEK293T cells were cultured in DMEM supplemented with 10% FBS, 100 U/ml Pen/Strep, 2 mM L-Glutamine, and 1 mM sodium

pyruvate. The cells were incubated at 37°C in 5% $CO_2$ and passaged whenever cells reached confluency for line maintenance. For biochemical assays, HEK293 cells were seeded at a density of 300,000 cells/well in 6-well plates or 1 million cells/10 $cm^3$ plate. 48 h later, a total of 3 μg of plasmid/well or 6 μg of plasmid/plate was transfected onto HEK293 cells. Plasmids were suspended in 420 μl Opti-MEM, and 3 μl X-tremeGENE HP DNA Transfection Reagent (#06366236001; Roche) per 1 μg plasmid was added. Transfected HEK293 cells were incubated at 37°C in 5% $CO_2$ for 48 h before cell lysis.

## Biochemical assays

### mRNA extraction and cDNA preparation

Cells stored in TRIzol (#15596026; Invitrogen) were brought to room temperature and resuspended in a final volume of 1 ml of TRIzol. 200 μl of chloroform was added to each sample and mixed thoroughly. Samples were centrifuged at 12,000 $g$ for 15 min at 4°C for phase separation and the clear aqueous phase was collected. 2 μl of GlycoBlue Coprecipitant (15 mg/ml, #AM9515; Invitrogen) and 500 μl of isopropanol were added to each sample, centrifuged at 12,000 $g$ for 10 min at 4°C, precipitating RNA as a blue pellet. The RNA pellet was rinsed in 75% ethanol, air-dried, and resuspended in 40 μl of nuclease-free water. RNA was isolated using the Zymo Research RNA Clean & Concentrator-5 Kit per manufacturer's protocol (#R1014; Zymo Research). mRNA concentration in each sample was quantified via Qubit RNA HS Assay Kit (#Q32852; Invitrogen). RNA samples were then diluted with nuclease-free water to match concentrations across all samples.

cDNA libraries were then generated by incubating the samples with qScript cDNA SuperMix (#101414-102; VWR) and nuclease-free water for 5 min at 25°C, 30 min at 42°C, and 5 min at 85°C. The resulting cDNA was then diluted threefold with nuclease-free water to a final volume of 30 μl per sample and stored at −80°C.

### Real-time qPCR

cDNA samples were plated on a 96-well qPCR plate and incubated with Fast SYBR Green Master Mix (#4385616; Applied Biosystems), nuclease-free water, and the forward and reverse primers of interest at a ratio of 5 μl SYBR: 3 μl water: 0.5 μl forward primer: 0.5 μl reverse primer: 1 μl sample. Each sample was plated two to four times to ensure technical replicates. A no-cDNA sample (water with primers and Master Mix) served as a negative control. Cycle threshold values were collected for each well and normalized to 18S as a housekeeping gene. The sequences of forward (F) and reverse (R) primers used (5′→ 3′) are:

Ctnnd2: (F) 5′-CGCGCTCCGACAACAAAA-3′ and (R) 5′-GAGAGC AGGCTAGGCCTTTCA-3′
Cdh2: (F) 5′-GGCCTTGCTTCAGGCGTC-3′ and (R) 5′-CCTTGA AATCTGCTGGCTCG-3′
Cdh10: (F) 5′-CTACCCAATTCAGAGCAGCAC-3′ and (R)5′-CAAAGAGGACTACTATAACCAGCA-3′
Cdh11: (F) 5′-CGCAACGGCCCCAGAAATTA-3′ and (R) 5′-ACT CACTGAGGACATCACGC-3′

Cdh20: (F) 5′-GCACTCCAATGCCAGAAGTTG-3′ and (R) 5′-TGT CTGAATGAAGCTTGCCG-3′
GFAP: (F) 5′-TACCAGGAGGCACTTGCTCG-3′ and (R) 5′-CCA CAGTCTTTACCACGATGTTCC-3′
NSE: (F) 5′-AGGAGAAGGCCTGCAACTG-3′ and (R) 5′-CTTCGC CAGACGTTCAGATCT-3′
18S: (F) 5′-GCAATTATTCCCCATGAACG-3′ and (R) 5′-GGCCTC ACTAAACCATCCAA-3′.

### Protein extraction

Cultured cells were rinsed in 1X PBS on ice, scrapped, and centrifuged at 18,000 $g$ for 5 min at 4°C. The cell pellet was resuspended on ice in lysis buffer (25 mM Hepes [#H0887; Sigma-Aldrich], 150 mM KCl, 1.5 mM $MgCl_2$, 10% glycerol, 1X Protease Inhibitor [#4693132001; Roche cOmplete Protease Inhibitor Cocktail], 0.5% NP-40 [#28324; Thermo Fisher Scientific]). The cell suspension was vortexed for 10 s and returned to ice for 10 min a total of three times to lyse the cells, then centrifuged at 18,000 $g$ for 10 min to pellet non-solubilized debris. The supernatant containing solubilized protein was collected. Protein concentration was measured by a Micro BCA assay (#23235; Thermo Fisher Scientific and denatured in 2X Laemmli Sample Buffer (#1610737; Bio-rad) containing 5% β-mercaptoethanol for 5 min at 95°C or 45 min at 45°C depending on the type of protein to be quantified by Western blot. Samples were stored at −80°C.

### Western blot

Protein samples were loaded onto a 4–15% Mini-PROTEAN TGX Stain-Free Protein Gel (#4568084; Bio-rad) and ran at 100 V for 100 min. The gel was then transferred to a polyvinylidene difluoride membrane at 100 V for 7 min using a Power Blotter (Invitrogen). Membranes were washed twice in distilled water and stained with a Total Protein Universal Stain (No-Stain Protein Labelling Reagent, #A44449; Invitrogen) for 10 min at room temperature, then washed twice in distilled water and imaged on a Universal gel imager (iBright; Thermo Fisher Scientific). Membranes were then washed once with 0.1% Tween20 in 1X TBS (0.1% TBST) and blocked for 1 h in 3% bovine serum albumin in 0.1% TBST. Membranes were incubated in primary antibody diluted 1:1,000 in 0.1% TBST overnight at 4°C. The next day, membranes were washed three times with 0.1% TBST, incubated in secondary antibody (1:5,000; LI-COR) for 2 h at room temperature, washed three times with 0.1% TBST, and imaged on an Odyssey Infrared Imaging System on the Image Studio software. The following primary antibodies were used: rabbit anti-δ-catenin (#ab184917; Abcam), mouse anti-β-actin (#A5541; Sigma-Aldrich), chicken anti-GFP (#GFP-1020; Aves), rabbit anti-GFAP (#Z033429-2; Dako), mouse anti-Tuj1 (#MAB1195; R&D systems), rabbit anti-β-tubulin (#926-42211; LI-COR), rat anti-HA (#11867423001; Sigma-Aldrich), and mouse anti-N-Cadherin (#610920; BD Biosciences).

### Co-IP assay

HEK 293T cells were grown, transfected, and lysed as described above. For each Co-IP condition, 1 million HEK 293T were plated in 10 $cm^3$ dishes. We prepared 30 μl of Pierce Anti-HA magnetic

beads (#88837; Thermo Fisher Scientific) per Co-IP condition. Beads were washed three times by adding 500 µl lysis buffer to the bead solution, standing on a magnetic tube rack for 2 min, and then removing the buffer for each wash. 100 µl lysis buffer per Co-IP condition was then added to the final solution of beads.

30–50 µg of suspended protein was taken for the input sample and denatured in 2X Laemmli Sample Buffer containing 5% β-mercaptoethanol for 5 min at 95°C. For the Co-IP assay, 100 µl of washed and resuspended anti-HA magnetic beads were added to 500–1,000 µg of cell suspension supernatant in lysis buffer (final volume 800 µl) and incubated rotating at 4°C for 2 h. Following incubation, Co-IP samples were placed on the magnetic rack for 2 min to separate unattached proteins from immunoprecipitated proteins. The beads were then washed three times with 500 µl lysis buffer each wash by flicking the tube, pulse spinning, and letting stand on the magnetic rack for 2 min. After three washes, the co-immunoprecipitated protein was eluted by incubating the beads in 60 µl 2X Laemmli Sample Buffer at 42°C for 15 min, then standing on the magnetic rack for 2 min. The protein eluate was saved and denatured in 2X Laemmli Sample Buffer containing 5% β-mercaptoethanol for 5 min at 95°C. Protein input samples and immunoprecipitated samples were run on a Western blot as described above.

### Surface biotinylation
HEK293T cells or purified astrocyte cultures were grown and transfected as described above. For each surface biotinylation condition, 600,000 cells were plated into two wells of a 6-well plate (300,000 cells per well). 48 h after transfection, HEK293T cells were washed three times on ice with PBS/Ca$^{2+}$/Mg$^{2+}$ (1 mM CaCl$_2$, 0.5 mM MgCl$_2$ in 1X PBS). Cells were then incubated on ice for 30 min in 0.5 mg/ml of EZ-Link Sulfo-NHS-LC-Biotin reagent (#A39257; Thermo Fisher Scientific) in PBS/Ca$^{2+}$/Mg$^{2+}$ and subsequently washed three times with ice-cold 1X TBS to quench the biotin reaction. Cells were then scraped and lysed in lysis buffer (1X radioimmunoprecipitation assay buffer [#R0278; Sigma-Aldrich: 50 mM Tris-HCl, pH 7.5, 150 mM NaCl, 1% NP-40, 0.5% sodium deoxycholate, 1 mM EDTA, 0.1% SDS, 0.01% sodium azide]) with 1X Protease Inhibitor and incubated rotating at 4°C for 40 min. The cell suspension was then centrifuged at 18,000 $g$ at 4°C for 5 min and the supernatant containing solubilized protein was collected.

60 µl of streptavidin beads per condition (Pierce High Capacity NeutrAvidin Agarose, #29202; Thermo Fisher Scientific) were washed three times in 1X PBS by rotating for 10 min and centrifuging at 8,000 $g$ for 3 min for each wash. Protein concentration was measured by Micro BCA assay. 120 µl of the lysis supernatant was saved as protein input sample and denatured in 2X Laemmli Sample Buffer containing 5% β-mercaptoethanol for 5 min at 95°C. 600 µl of the lysis supernatant was added to a fresh 1.5 ml tube along with 60 µl of washed NeutrAvidin beads and incubated rotating at 4°C for 2 h.

Following incubation, samples were centrifuged 8,000 $g$ for 3 min. The NeutrAvidin beads were rinsed three times ice-cold 1X PBS and once in lysis buffer by rotating for 5 min and centrifuging for 3 min at 8,000 $g$ per wash. After the final wash, 60 µl of elution buffer (1.2 mg of biotin [#B4501; Sigma-Aldrich],

4% SDS, 20% glycerol, 0.1% β-mercaptoethanol, 125 mM Tris, pH 6.8; 20 ml in keratin-free water) was added to the beads and the biotinylated protein was eluted by heating for 15 min at 60°C, then centrifuging at 8,000 $g$ for 1 min to pellet the beads. The biotinylated protein eluate was saved and denatured in 2X Laemmli Sample Buffer containing 5% β-mercaptoethanol for 5 min at 95°C. Protein input samples and biotinylated pulldown samples were run on a Western blot as described above.

### Protein interaction modeling
To model the δ-catenin:N-cadherin interaction, full-length mouse N-Cadherin and the ARM domain (533–1,018 aa) of human δ-catenin were predicted and docked using Alphafold 2.0 Multimer. The highest-confidence structure, calculated via predicted local distance difference test, was then energy-minimized using Amber relaxation. The resulting minimized structure was then used to generate representative models in the PyMOL molecular visualization system.

To model the electrostatic impacts of mutations in the δ-catenin ARM domain, wild-type and mutant δ-catenin ARM domain structures were generated and energy-minimized using Alphafold 2.0 Multimer, as described above. UCSF ChimeraX molecular visualization software was used to generate and visualize the electrostatic properties of the resulting δ-catenin ARM domain structures.

### Surgical procedures
#### PALE
Late P0/early P1 CD1 mouse pups were sedated by hypothermia until anesthetized. 1 µl of plasmid DNA (1 µg of pGLAST-PBase and 1 µg of pPB-shRNA-mCherryCAAX, mixed with FastGreen Dye) was injected into the lateral ventricles of one (for PALE experiments only) or both hemispheres (for IUE/PALE experiments) using a mouth pipette (Drummond Scientific). Following DNA injection, electrodes were oriented with the positive terminal above the cortex and the negative terminal below the chin, and five 50-ms pulses of 100 V spaced 950 ms apart were applied. Pups recovered from sedation on a heating pad, returned to their home cage for monitoring, and collected at P7, P14, and/or P21. Experimental groups were randomly assigned within each litter. All animals that appeared healthy at the time of collection were processed for data collection, and all brains were examined for the presence of electroporated cells using immunohistochemistry. Brains that did not have any successful labeling of astrocytes or properly formed cortices due to errors in the procedure were excluded from the study.

#### IUE
Timed-pregnant CD1 mice (E15.5) were anesthetized using isoflurane and were subjected to laparotomy. To expose the uterine horns, a small incision was made in the medial ventral abdomen of the pregnant mouse, and the uterus, along with the embryos, was gently pulled out of the abdominal cavity. To visualize the embryos through the uterus, a fiber optic light source was used, and plasmids at a concentration of 1 µg/µl were injected into the brain's lateral ventricle using pulled glass capillaries. 0.1% FastGreen was used to visualize the injected area of the plasmid

solution. Using a 3-mm Tweezer electrode, all the embryos on both sides of the uterus were electroporated with five 50-ms square pulses of 40 V with 999-ms intervals generated from a BTX ECM830 electroporator. After electroporation, the uterine horns were placed inside the abdomen, and the abdominal cavity was filled with normal saline. The incision was closed by suturing the muscle wall and skin, and the mouse was allowed to recover on a heating pad.

**Immunocytochemistry**
Astrocyte–neuron co-cultures, astrocyte monocultures, astrocyte mosaic cultures, and cultures assessing neuron morphology were fixed with warm 4% paraformaldehyde (PFA) for 7 min at room temperature. Cells were then washed with DPBS, blocked in an antibody blocking buffer containing 50% normal goat serum (NGS; #16210-064; Gibco) in 0.4% Triton X-100 Surfact-Amps Detergent Solution (#28314; Thermo Fisher Scientific) for 30 min, and incubated in primary antibodies diluted 1:1,000 in blocking buffer overnight at 4°C. Samples were then washed with DPBS, incubated in Alexa Fluor conjugated secondary antibodies diluted 1:1,000 for 2 h at room temperature, and finally washed with DPBS.

The following combinations of primary and secondary antibodies were used for the respective culture assays. Astrocyte–neuron co-cultures and astrocyte monocultures were stained with chicken anti-GFP and rabbit anti-GFAP or rat anti-HA primary antibodies before being stained with Alexa Fluor 488 goat anti-chicken IgY(H+L) (#A-11039; Invitrogen) and Alexa Fluor 594 goat anti-rabbit IgG(H+L) (#A-11037; Invitrogen) or Alexa Fluor 594 goat anti-rat IgG(H+L) (#A-11007; Invitrogen). Astrocyte mosaic cultures were stained with chicken anti-GFP and rabbit anti-RFP (#600-401-379; Rockland) primary antibodies before being stained with Alexa Fluor 488 goat anti-chicken IgY(H+L) and Alexa Fluor 594 goat anti-rabbit IgG(H+L). Cultures assessing neuronal morphology were stained with chicken anti-GFP and mouse anti-Tuj1 antibodies before being stained with Alexa Fluor 488 goat anti-chicken IgY(H+L) and Alexa Fluor 594 goat anti-mouse IgG(H+L) (#A-11032; Invitrogen).

Coverslips were mounted onto glass slides with VECTA-SHIELD Antifade Mounting Medium with DAPI (#H-1200-10; Vector Laboratories) and imaged at 21°C on an Axioimager M1 (Zeiss) fluorescent microscope with an EC Plan-Neofluar 40× oil NA 1.3 objective (Immersol 518F) and an AxioCAM MRm camera, supported by the AxioVision SE64 software.

GFP-positive astrocytes meeting the following criteria were imaged: (1) grown on healthy, evenly distributed neurons or astrocytes, (2) robust expression of fluorescent markers, (3) containing a single nucleus as revealed by DAPI stain, and (4) non-overlapping with other GFP expressing astrocytes. The total number of astrocytes imaged per condition is reflected in figure legends, but generally, ~30 astrocytes from three different coverslips were imaged per condition within an experiment. Healthy Tuj1-positive neurons grown on healthy astrocytes strongly expressing GFP were similarly imaged. A total of 20 neurons from three different coverslips were imaged per condition within an experiment. A total of three to four experiments were conducted for each assay.

Astrocyte or neuron morphological complexity was analyzed in FIJI using Sholl analysis plugin (Ferreira et al., 2014). Image acquisition and analysis were performed blinded to experimental conditions. Each independent experiment consisted of primary neurons and astrocytes isolated from a unique litter of wild-type mice of mixed sexes. The peak number of intersections of control astrocytes indicated health of primary cultures. Therefore, if Sholl analysis indicated that control astrocytes had a peak intersection of <20 for astrocyte–neuron co-cultures and a peak intersection of <6 for astrocyte cultures, the results of that entire experiment were excluded from analysis. Sholl analysis curves from three to four independent experiments were statistically compared using co-variance with a Tukey's post-hoc test. The R script used for graphing and statistical analysis can be found at: https://github.com/Eroglu-Lab/Sholl_Macro. The total astrocyte process was measured in ImageJ using the NeuroAnatomy SNT plugin (Arshadi et al., 2021) and analyzed using nested one-way ANOVA with a Tukey's post-hoc test on Graphpad Prism 9. Data distribution was assumed to be normal based on QQ plots but this was not formally tested.

**Immunohistochemistry**
Mice were anesthetized by injection of 200 mg/kg tribromoethanol (avertin) and perfused with 1X TBS and 4% PFA. Brains were collected and post-fixed in 4% PFA overnight, then cryoprotected in 30% sucrose. Brains were frozen in a 2:1 solution of 30% sucrose/Tissue-Tek O.C.T. Compound (#15710; Electron Microscopy Sciences) and stored at –80°C until time of sectioning. For PALE experiments, coronal tissue sections of 100 µM thickness were collected and stored in a 1:1 solution of TBS/glycerol at –20°C. For all other experiments, 40 µM sections were collected and stored in the same manner.

For immunostaining in PALE experiments, 100 µM floating sections were washed and permeabilized in 1X TBS containing 0.4% Triton X-100 (0.4 % TBST), blocked in 10% NGS diluted in 0.4% TBST, and incubated shaking in the following primary antibodies for three nights at 4°C: rat anti-Ctip2 (1:500, #Ab00616-7.4; Absolute Antibody), rabbit anti-RFP (1:2,000), and mouse anti-Satb2 (1:500, #ab51502; Abcam). Following primary incubation, sections were washed in 0.4% TBST and incubated shaking in Alexa Fluor 488 goat anti-rat IgG(H+L) (#A-11006; Invitrogen), Alexa Fluor 568 goat anti-rabbit IgG(H+L), and Alexa Fluor 647 goat anti-mouse IgG(H+L) (#A-21236; Invitrogen) secondary antibodies diluted 1:100 for 3 h at room temperature. Sections were then washed with 0.4% TBST and mounted onto glass slides using a mounting media (STED mounting media recipe). For DAPI nuclear marker staining, DAPI (#D1306; Invitrogen) was added to each well during the final 30 min of secondary incubation at a concentration of 1:50,000. IUE/PALE brains were sectioned similarly. These sections were stained with chicken anti-GFP (1:1,000), rabbit anti-RFP (1:2,000), and rat anti-Ctip2 (1:500) primary antibodies followed by Alexa Fluor 488 goat anti-chicken IgY(H+L), Alexa Fluor 594 goat anti-rabbit IgG(H+L), and Alexa Fluor 647 goat anti-rat IgG(H+L) (#A-21247; Invitrogen) secondary antibodies diluted 1:100.

For immunostaining related to characterization of PALE knockdown astrocytes, 40 μM sections were washed and permeabilized in 1X TBS containing 0.2% Triton X-100 (0.2% TBST), blocked in 10% NGS diluted in 0.2% TBST, and incubated shaking in primary antibody for overnight at 4°C. To quantify the extent of δ-catenin knockdown, sections were stained with rabbit anti-δ-catenin (1:500) and chicken anti-RFP (1:1,000, #600-901-379; Rockland). Another set of 40-μM sections were stained with rabbit anti-PSD95 (1:300, #51-6900; Invitrogen), chicken anti-RFP (1:1,000), and guinea pig anti-bassoon (1:1,000, #141318; Synaptic systems). Following primary incubation, sections were washed in 0.2% TBST and incubated shaking in Alexa Fluor conjugated secondary antibodies diluted 1:200 for 2 h at room temperature. The following secondary antibodies were used: Alexa Fluor 488 goat anti-rabbit IgG(H+L) (#A-11034; Invitrogen), Alexa Fluor 594 goat anti-chicken IgY(H+L) (#A-11042; Invitrogen), Alexa Fluor Plus 405 goat anti-rabbit IgG(H+L) (#A48254A48254; Thermo Fisher Scientific), and Alexa Fluor 647 goat anti-guinea pig IgG(H+L) (#A-21450; Invitrogen). Sections were then washed with 0.2% TBST and mounted onto glass slides using STED mounting media. For DAPI nuclear marker staining, DAPI (#D1306; Invitrogen) was added to each well during the final 10 min of secondary incubation at a concentration of 1:50,000. Images were acquired at 21°C on an Olympus FV 3000 microscope with an Olympus Plan APO 60× 1.4 NA oil objective (Immoil-f30cc), supported by the Olympus FV315-SW software. The researcher acquiring images was blinded to the experimental group.

### Whole astrocyte Sholl analysis and territory volume analysis of PALE experiments

Whole astrocytes in the primary visual cortex were imaged at high magnification (60× objective, 2× zoom) and resolution (0.5 μM step size, 50–60 μM z-stack) on an Olympus FV 3000 microscope. Imaged astrocytes were then reconstructed in 3D using a filament tracer on Imaris Bitplane 9.9. Two custom Xtensions were developed with Imaris. "3D Sholl analysis" was used to quantify the complexity of the reconstructed astrocyte processes. The number of intersections every 5 μM from the cell soma was calculated using this extension as a measure of 3D Sholl analysis. Astrocyte complexity by 3D Sholl analysis was analyzed across experimental conditions using one-way ANOVA and a linear mixed model followed by Tukey post-hoc test. The R script used for graphing and statistical analysis can be found at https://github.com/Eroglu-Lab/In-vivo-Sholl-Analysis. "Convex Hull" was used to create a surface that represented the territory of the astrocyte. Astrocyte territory volume across experimental conditions was analyzed using nested *t* test in Graphpad Prism 9. The exact number of astrocytes and animals analyzed in each condition is specified in the figure legend, but at least three whole astrocytes were imaged per animal, and six to nine mice of both sexes were used for each experimental condition. Data distribution was assumed to be normal based on QQ plots but this was not formally tested.

### Synapse analysis of PALE knockdown astrocytes

Regions of astrocytes in the primary visual cortex were imaged at high magnification (60× objective, 1.64× zoom) and resolution (0.34 μM step size, 3.06 μM z-stack) on an Olympus FV 3000 microscope. SynBot, an ImageJ-based software developed by the Eroglu lab (Savage et al., 2023 *Preprint*) was utilized to quantify synapses using the following parameters: Each image was subdivided into three 1.02-μM substacks, z-projected, and the synaptic puncta identified by manual thresholding and pixel-based colocalization. The synaptic puncta were normalized to the area of astrocyte to give synaptic density. All scripts related to Synbot specific to this manuscript can be found at https://github.com/Eroglu-Lab/Syn_Bot/releases/tag/v0.2.0. Data distribution was assumed to be normal based on QQ plots but this was not formally tested.

### Multiplexed immunofluorescence and RNA-FISH for simultaneous detection of protein and RNA targets

Four to six 20-μM sections of primary visual cortex from P1, P7, P14, and P21 Aldh1L1-EGFP were directly mounted onto glass slides. All sections were kept at –80°C until use, and only sections <1 mo in age were utilized. Custom-designed probes for Ctnnd2 were purchased from Molecular Instruments utilizing their HCR platform and prepared per the manufacturer's protocols. First, sections were washed three times in 1X PBS containing 0.1% Tween20 (PBST), blocked in 200 μl of antibody buffer at room temperature for 1 h, and incubated with chicken anti-GFP (1:1,000) primary antibody overnight at 4°C. The sections were washed with PBST before application of 100 μl initiator-labeled secondary antibody diluted in antibody buffer. After 1 h incubation at room temperature, the sections were washed in PBST, postfixed in 200 μl of 4% PFA for 10 min before immersion in 5× sodium chloride sodium citrate containing 0.1% Tween20 (5× SSCT) for 5 min to prime the sections for RNA probe hybridization. 100 μl of 16 nM of Ctnnd2 probe solution was applied to the sections before incubation overnight in a 37°C humidified chamber. To remove excess probes, the sections were incubated in a mixture of probe wash buffer and 5× SSCT at 37°C. Each incubation step was 15 min long and had an increasing concentration of 5× SSCT, ending with a final incubation of 100% 5× SSCT at room temperature. Next, 200 μl of amplification buffer was added to the sections for 30 min at room temperature to prepare the sections for probe amplification. Snap-cooled hairpins specific to the initiator-labeled secondary antibody and Ctnnd2 probe were added to amplification buffers resulting in a 60 mM hairpin solution. The sections were incubated overnight with 100 μl of the hairpin solution at room temperature. Finally, the sections were immersed in 5× SSCT for a total of 45 min to remove excess hairpins, dried, and mounted with STED mounting media.

The resulting sections were imaged within 2 d at high magnification (60× objective) and resolution (1 μM step size, 5 μM z-stack) on an Olympus FV 3000 microscope. In total, 180 images were taken for the entire dataset (images of L1, L2/3, L4, L5, and L6 of each section, three sections per animal, three animals per time point for P1, P7, P14, and P21). The images were then processed using a custom pipeline that includes thresholding of Ctnnd2 signal and segmentation of astrocyte soma using Ilastik. All macros and R scripts related to this pipeline can be found at https://github.com/Eroglu-Lab/RNA-FISH-analysis-pipeline.

Ctnnd2 mRNA expression is defined as the volume of Ctnnd2 mRNA normalized to the volume of cell soma and analyzed across developmental timepoints and cortical layers using two-way ANOVA, followed by Dunnett's multiple comparisons test. Data distribution was assumed to be normal but this was not formally tested.

### Quantification and statistical analysis
All statistical analyses were performed using Graphpad Prism 9, except for astrocyte 2D and 3D Sholl analysis, which we analyzed using a custom R code. Data distribution was assumed to be normal, but this was not formally tested. The exact number of replicates, specific statistical tests, and P values for each experiment are indicated in the figure legends. All data are represented as mean ± SE of the mean, and individual data points are shown for all data, where applicable. Specific details for inclusion, exclusion, and randomization are included in the specific subsections of the Materials and methods details section.

### Online supplemental material
Fig. S1 contains qPCR and Western blot validation that astrocytes express Ctnnd2/δ-catenin in neuron and astrocyte cultures. Fig. S2 emphasizes how silencing astrocytic δ-catenin expression reduces astrocyte morphological complexity only in astrocyte–neuron co-cultures. Fig. S3 shows how silencing astrocyte-enriched cadherins reduces astrocyte morphological complexity. Fig. S4 validates the PALE methodology of silencing Ctnnd2 expression specifically in astrocytes in vivo. Fig. S5 is a graphical representation of cadherin expression in P14 cortical astrocytes and neurons based on single-cell transcriptomic datasets.

### Data availability
The data are available from the corresponding author upon reasonable request.

## Acknowledgments
We thank Drs. Jeff Stogsdill, Isabella Tselnicker, Paola Arlotta, and Nicola Allen. We thank Dr. Matthew J. Gastinger from Imaris for his assistance in developing the custom Xtensions described in this paper. We thank Dr. Trisha Vaidyanathan, Pia Rodriguez, and other Eroglu lab members for the critical review of the manuscript.

This work was supported by the National Institutes of Health (R01 NS102237 and R01 DA047258 to C. Eroglu, F32 NS112565 for K. Sakers, and F31NS125985 for J. Ramirez) and Simons Foundation Autism Research Initiative Pilot Grant (ID 400729 to C. Eroglu). C. Eroglu is a Howard Hughes Medical Institute Investigator.

Author contributions: Conceptualization: C.X. Tan and C. Eroglu; Investigation: C.X. Tan, D.S. Bindu, and E. Hardin; Software: C.X. Tan, K. Sakers, J. Ramirez, and J. Savage; Methodology: C.X. Tan, C. Eroglu, and R. Baumert; Data curation: C.X. Tan and J. Savage; Formal analysis: C.X. Tan and C. Eroglu; Writing—original draft: C.X. Tan and E. Hardin; Writing—Original Visualization: C.X. Tan; Writing—review & editing: C.X. Tan, C. Eroglu, D.S. Bindu, E. Hardin, K. Sakers, R. Baumert, J. Ramirez, and J. Savage; Project administration: C.X. Tan; Supervision: C. Eroglu; Funding acquisition: C. Eroglu, K. Sakers, and J. Ramirez.

Disclosures: The authors declare no competing interests exist.

Submitted: 29 March 2023

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

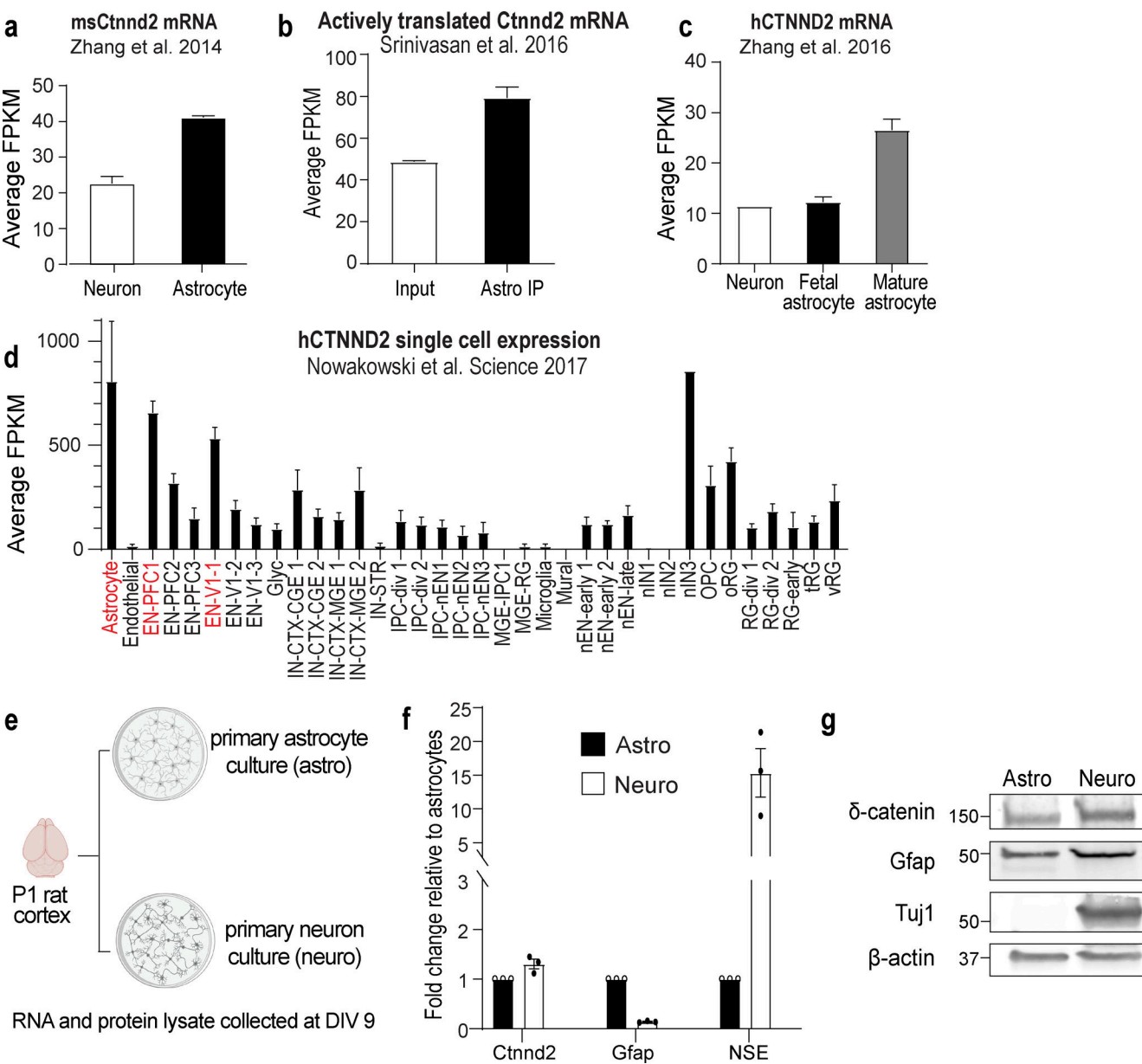

Figure S1. **Ctnnd2 is expressed in astrocytes. (a)** *Ctnnd2* mRNA is more highly expressed in P7 purified mouse astrocytes relative to P7 purified mouse neurons. Average FPKM of *Ctnnd2* mRNA from Zhang et al. (2014). **(b)** *Ctnnd2* mRNA is actively transcribed. Average FPKM values of *Ctnnd2* mRNA engaged within translating astrocyte-specific and all ribosomes (input) from Srinivasan et al. (2016). Astrocyte-specific mRNA were isolated from P80 Aldh1L1–Cre[ERT2]/Ribo-tag mice. **(c)** *CTNND2* mRNA is most highly expressed in mature human astrocytes. Average FPKM values of *CTNND2* mRNA derived from human neurons, fetal astrocytes, and mature astrocytes, as described in Zhang et al. (2016). **(d)** In the developing human fetal brain, *CTNND2* mRNA is abundant in astrocytes and excitatory neurons in the V1 visual cortex and prefrontal cortex. Average transcripts per million of *CTNND2* mRNA in each cell type identified in single-cell transcriptomic analysis of the developing human telencephalon from Nowakowski et al. (2017). **(e)** Schematic of isolating primary astrocyte culture (astro) and primary neuron culture (neuro) from P1 rat cortex. **(f)** Comparable expression of *Ctnnd2* mRNA transcripts by RT-PCR in DIV 9 primary astrocyte culture (astro) and primary neuron culture (neuro), P = 0.28. Specific primer sets were used to detect glial fibrillary acid protein (Gfap, astrocyte control) and neuron-specific enolase (NSE, neuron control). *n* = 3 independent cultures. Paired *t* test. **(g)** Western blot detection of δ-catenin in DIV 9 primary astrocyte culture (astro) and primary neuron culture (neuro). From f and g, purified astrocyte culture does not contain detectible neuron contamination (NSE in f and Tuj1 in g). All protein molecular weights are expressed in kiloDaltons (kD). All data are presented as mean ± SEM.

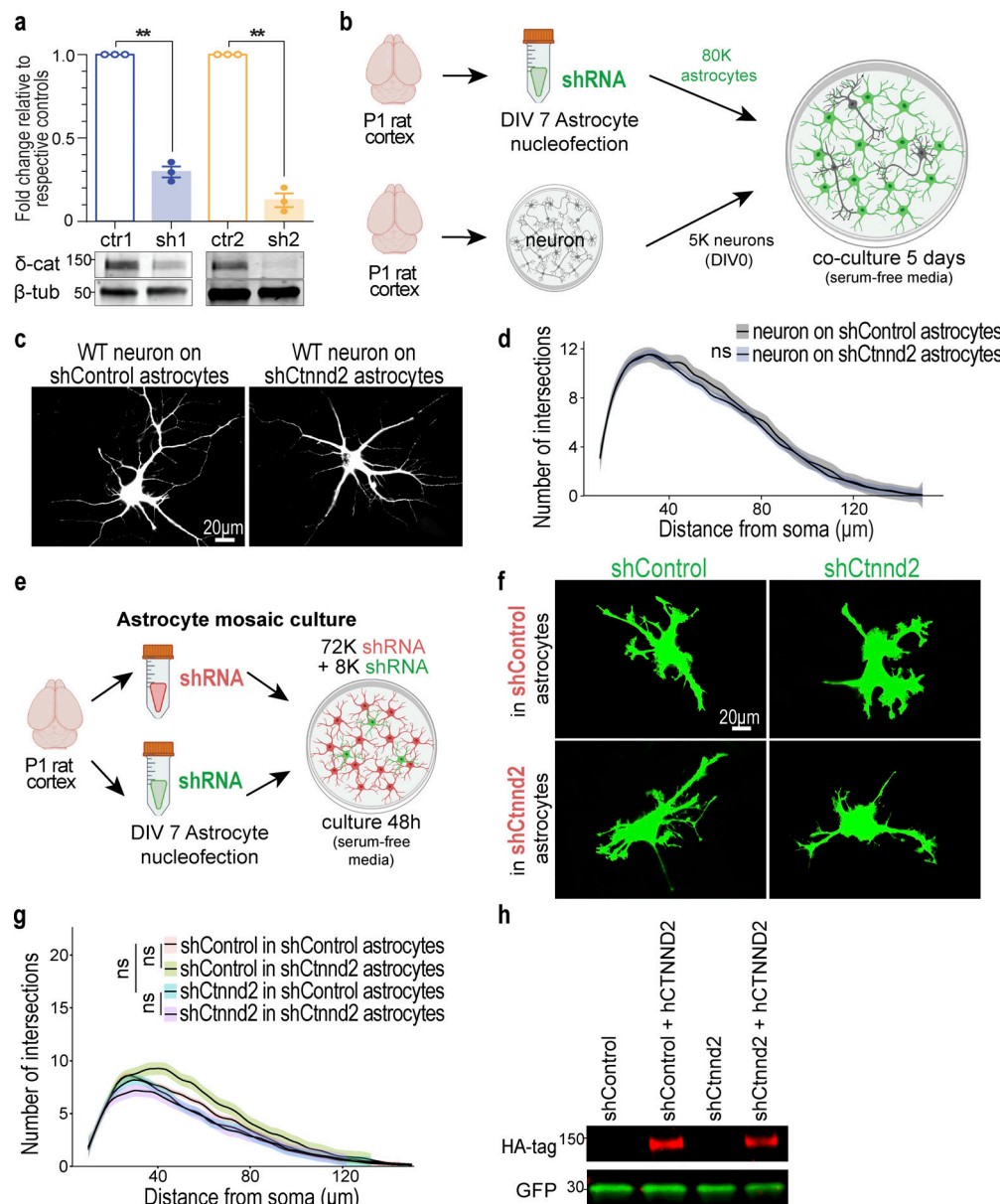

**Figure S2. Astrocytic δ-catenin does not regulate neuron morphology or astrocyte–astrocyte interactions. (a)** Two shRNAs targeting rat/mouse *Ctnnd2* were generated: shCtnnd2-1 (sh1) and shCtnnd2-2 (sh2). Both shRNAs effectively knock down δ-catenin (δ-cat) expression in astrocytes compared to their respective controls in which the shRNA sequence was scrambled (shControl-1, crt-1, and shControl-2, crt-2). Loading control: β-tubulin (β-tub). *n* = 3 independent experiments. Unpaired *t* test with Welch's correction. ** P < 0.01. **(b)** Schematic of neuron morphology assay. Astrocytes were isolated from P1 rat cortices to obtain purified astrocyte cultures. DIV 7 astrocytes are nucleoporated with pLKO.1 plasmid expressing hU6-shRNA and CAG-EGFP and seeded at a density of 80,000 astrocytes per coverslip. Neurons were isolated from another set of P1 rat cortices when astrocytes were DIV 7 and co-cultured on top of nucleoporated astrocytes for 5 d in NGM plus to allow for neurite growth. **(c)** Representative images of Tuj1-positive rat neurons after 5 d co-culture with shControl or shCtnnd2 nucleoporated astrocytes (not labeled). **(d)** Neuron morphology is unaffected by silencing of astrocytic Ctnnd2 in astrocyte–neuron co-culture paradigm (P = 0.58). Neuron morphology is quantified by Sholl analysis. *n* = 60 neurons per condition from three independent experiments. Linear mixed model with Tukey HSD. **(e)** Schematic of astrocyte mosaic culture. Astrocytes were isolated from P1 rat cortices to obtain purified astrocyte cultures. DIV 7 astrocytes are nucleoporated with pLKO.1 plasmid expressing hU6-shRNA and CAG-EGFP or a pPB-shRNA-mCherryCAAX plasmid. Nucleoporated astrocytes were seeded onto coverslips at a total density of 80,000 astrocytes per coverslip in a 9:1 ratio of pPB-shRNA-mCherryCAAX:pLOK.1-shRNA-EGFP for 48 h in NGM Plus. **(f)** Representative images of rat astrocytes nucleoporated with shControl-GFP or shCtnnd2-GFP after 48 h co-culture with astrocytes nucleoporated with shControl-mCherry or shCtnnd2-mCherry (not labeled). **(g)** Reduction in astrocyte morphology following δ-catenin knockdown in astrocyte–neuron co-culture assay is unrelated to defective astrocyte–astrocyte adhesion. Silencing Ctnnd2 did not influence wild-type astrocyte morphology (P = 0.18). Wild-type astrocyte morphology was unchanged when cultured with either shControl or shCtnnd2 astrocytes (P = 0.06). The same was also observed in shCtnnd2 astrocytes cultured with shControl or shCtnnd2 astrocytes (P = 0.91). Astrocyte morphology is quantified by Sholl analysis. *n* = 90 neurons per condition from three independent experiments. Linear mixed model with Tukey HSD. **(h)** Full-length human CTNND2 is insensitive to shCtnnd2. Western blot of HEK293 cell lysates transfected with shControl or shCtnnd2 in the presence or absence of HA-tagged, full-length human CTNND2 (hCTNND2). hCTNND2 expression was detected using an antibody against HA-tag. GFP expression denotes the presence of the shRNA construct. All protein molecular weights are expressed in kiloDaltons (kD). All data is presented as mean ± SEM. Scale bars: 20 μm.

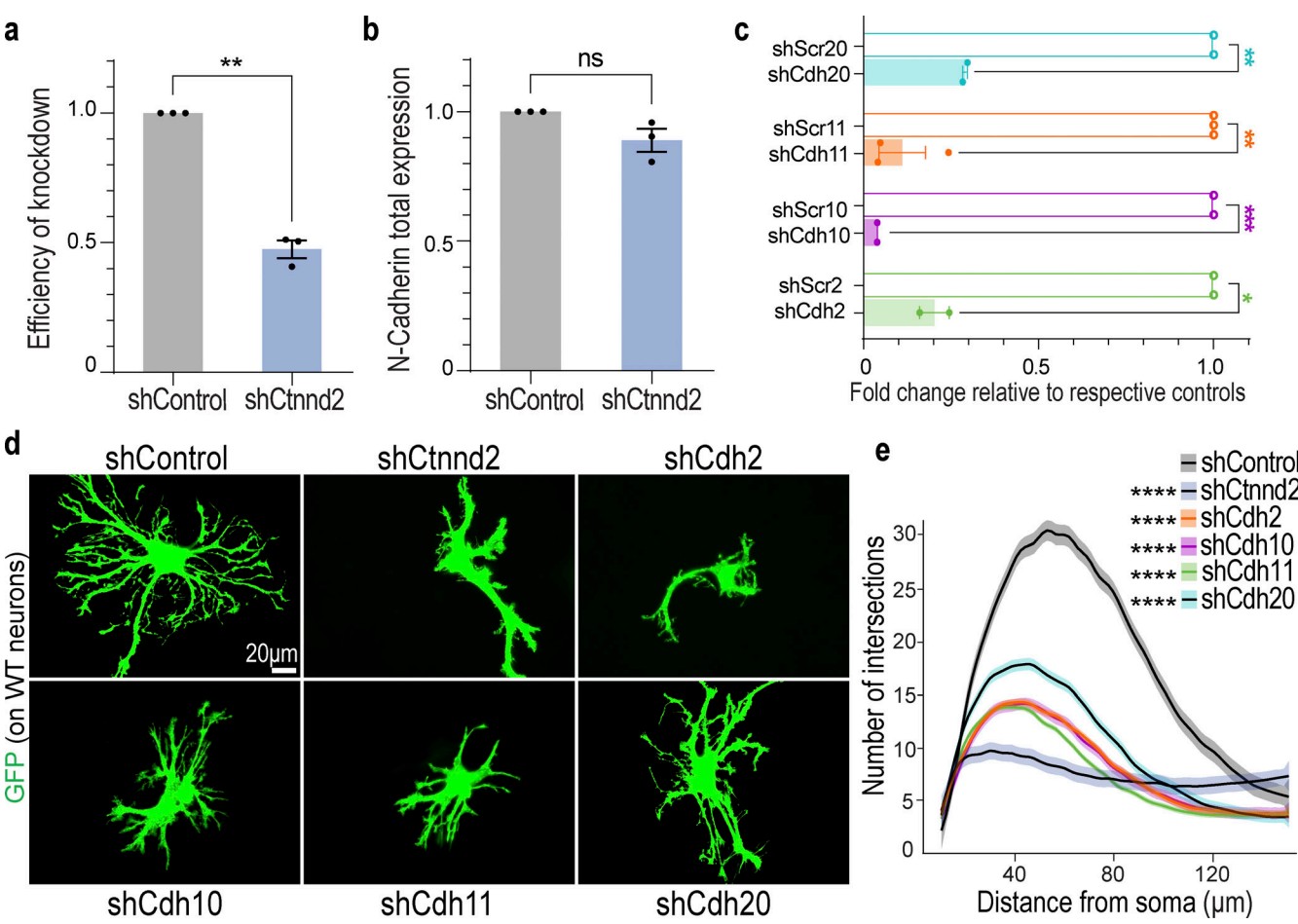

Figure S3. **Knockdown of astrocyte-enriched cadherins reduces astrocyte morphological complexity. (a)** Ctnnd2 knockdown efficiency in astrocyte surface biotinylation assay was 52.5 ± 0.03% (P = 0.004). $n$ = 3 independent replicates. Unpaired $t$ test with Welch's correction. **(b)** Silencing astrocytic Ctnnd2 did not alter total N-cadherin expression (P = 0.13). $n$ = 3 independent replicates. Unpaired $t$ test with Welch's correction. **(c)** shRNA targeting rat/mouse astrocyte-enriched cadherins were generated: shCdh2, shCdh10, shCdh11, and shCdh20. All four shRNAs effectively silenced gene target expression in astrocytes compared to their respective controls in which the shRNA sequence was scrambled (shScrambleCdh2, shScr2; shScrambleCdh10, shScr10; shScrambleCdh11, shScr11; and shScrambleCdh20, shScr20). Cadherin cDNA levels were quantified from transduced cortical astrocytes using RT-PCR. $n$ = 2–3 independent experiments. Unpaired $t$-test with Welch's correction. **(d)** Representative images of rat astrocytes (green) transfected with shControl, shCtnnd2-1, or shRNA constructs targeting astrocyte-enriched classical cadherins: Cdh2 (shCdh2), Cdh10 (shCdh10), Cdh11 (shCdh11), and Cdh20 (shCdh20) after 48 h co-culture with wild-type neurons (not labeled). **(e)** Silencing astrocyte-enriched cadherins impairs neuron-contact-dependent astrocyte morphogenesis. Knockdown of Cdh2 (P = $1.1 \times 10^{-16}$), Cdh10 (P = $1.1 \times 10^{-16}$), Cdh11 (P < $2.2 \times 10^{-16}$), and Cdh20 (P = $2.83 \times 10^{-12}$) individually in astrocytes significantly reduces astrocyte complexity. Quantification of astrocyte complexity by Sholl analysis. Average of 154 astrocytes per condition from three independent experiments. Linear mixed model with Tukey HSD. All data is presented as mean ± SEM. Scale bars: 20 µm. ** P < 0.01; **** P < 0.0001.

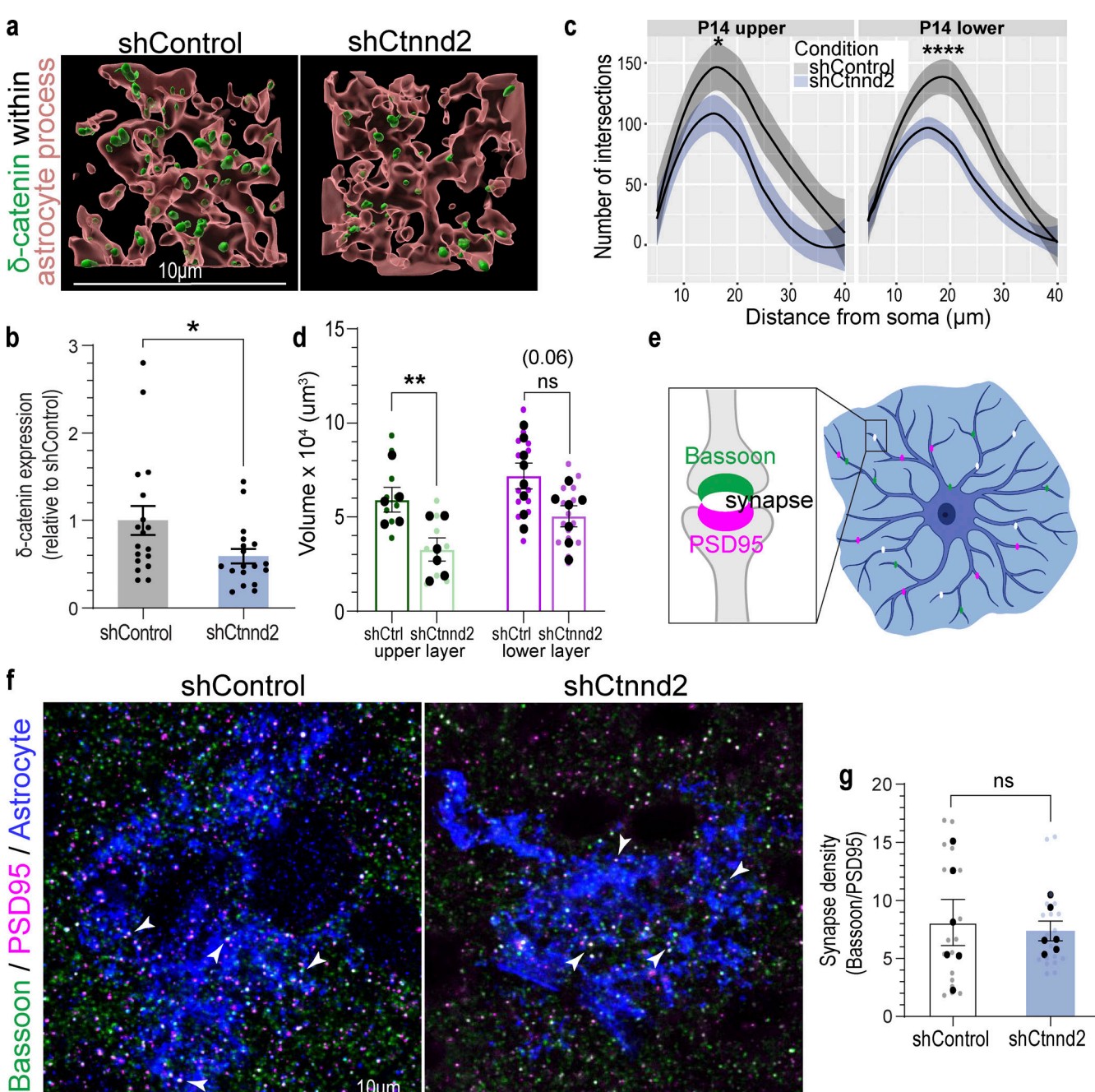

Figure S4. **Validation of PALE methodology. (a)** Representative images of δ-catenin (green) knockdown in P14 shControl and shCtnnd2 PALE astrocyte processes. **(b)** Silencing Ctnnd2 expression with shRNA decreased δ-catenin by 0.41-fold (P = 0.031). *n* = 18 astrocytes from five to seven animals per condition. Three regions of interest were analyzed for each astrocyte. Mann–Whitney U test because samples did not pass Shapiro–Wilk normality test. **(c)** Silencing astrocytic δ-catenin resulted in a significant decrease in astrocyte complexity in P14 upper-layer (P = 0.015) and P14 lower-layer (P = 4.6 × 10$^{-4}$) astrocytes. Quantification of in vivo astrocyte complexity with 3D Sholl analysis. *n* = 24–25 astrocytes from seven to eight mice. One-way ANOVA, linear mixed model with Tukey HSD. **(d)** Silencing astrocytic δ-catenin resulted in a significant decrease in astrocyte territory volume in P14 upper-layer astrocytes (P = 0.0015), but not P14 lower-layer astrocytes (P = 0.06). Quantification of in vivo astrocyte territory volumes by convex hull analysis. Only astrocytes from V1 cortex were imaged and analyzed. The average astrocyte territory volume of individual mice is plotted in black. *n* = 24–45 astrocytes from seven to eight mice, nested *t* test for each time point. ns, not significant. **(e)** Schematic of quantifying synaptic density within astrocyte. Excitatory synapses are identified by the colocalization of Bassoon (presynaptic marker, green) and PSD95 (postsynaptic marker, magenta). Synapses within the astrocyte (blue) are quantified and normalized to the area of the astrocyte (blue outline) to obtain synaptic density. **(f)** Representative images of P14 shControl and shCtnnd2 PALE astrocytes (blue) stained with Bassoon (green) and PSD95 (magenta). Excitatory synapses are identified by the colocalization of Bassoon and PSD95 (white, denoted by arrowheads). **(g)** PALE knockdown of Ctnnd2 did not affect density of excitatory synapses within astrocyte neuropil at P14 (P = 0.72). Quantification of excitatory synapse density by counting colocalized Basson/PSD95 puncta within astrocyte normalized to area of astrocyte. Only astrocytes from V1 cortex were imaged and analyzed. The average synapse density of individual mice is plotted in black. *n* = 19–20 astrocytes from six mice, nested *t* test. ns, not significant. All data is presented as mean ± SEM. Scale bars: 10 μm. * P < 0.05; ** P < 0.01.

| | | Type 1 | | Type II specificity group A | | | Type II specificity group B | | | | Type II specificity group C | |
|---|---|---|---|---|---|---|---|---|---|---|---|---|
| | | Cdh2 | Cdh4 | Cdh6 | Cdh9 | Cdh10 | Cdh7 | Cdh12 | Cdh18 | Cdh20 | Cdh8 | Cdh11 |
| **astrocyte** Farhy-Tselnicker et al. 2021 | layer | ● (green, large) | ● (green, small) | | | ● (green, large) | | | ● (green) | ● (green) | | ● (green, small) |
| | lower layer | ● (magenta) | ● (magenta, small) | ● (magenta, small) | | ● (magenta, large) | | | ● (magenta) | ● (magenta) | | ● (magenta) |
| **neuron** Stogsdill et al. 2022 | L2/3 | | *Inhibitory* | | | ■ (green, small) | *Inhibitory* | ■ (green, small) | | | | |
| | L4 | | | | | | | ■ (green) | | | | |
| | L5 | ■ (small) | | ■ (magenta, small) | | ■ (magenta) | | ■ (magenta) | ■ (magenta) | ■ (magenta, small) | ■ (magenta) | ■ (magenta) |
| | L6 | ■ (magenta) | | | ■ (magenta, small) | ■ (magenta) | | ■ (magenta) | ■ (magenta, large) | | | ■ (magenta) |

**Legend**

Astrocyte (FPKM)
- 10 – 30
- 31 – 50
- 51 – 100
- 101 – 500
- >501

Neuron (FPKM)
- 10 – 20
- 21 – 30
- 31 – 40
- 41 – 60
- >100

- upper layer (green)
- lower layer (magenta)

Figure S5. **Cadherin expression in P14 cortical astrocytes and cortical neurons.** Layer-specific expression of Type I and Type II classical cadherins were obtained from single-cell transcriptomic datasets of P14 cortical astrocytes (Farhy-Tselnicker et al., 2021) and neurons (Stogsdill et al., 2022). In both datasets, cadherins with FPKM values <10 were discarded. In the original astrocyte dataset, upper-layer astrocyte populations were subdivided into two groups corresponding to L1 and L2–L4. Our analysis did not consider cadherin expression in L1 cortical astrocytes to match the neuronal dataset. In our neuronal dataset, we only considered excitatory neurons and grouped by layers. However, we noted that Cdh4 and Cdh7 were more highly expressed in inhibitory neurons. From the 18 cadherins, there were 11 cadherins with significant mRNA expression in the P14 cortex. We plotted their relative expressions, grouping cadherins according to their binding affinities.

