## [Peer Review File · The Journal of Cell Biology]

δ -catenin controls astrocyte morphogenesis via layer-specific astrocyte-neuron cadherin interactions

Christabel Tan, Dhanesh Sivadasan Bindu, Evelyn Hardin, Kristina Sakers, Ryan Baumert, Juan Ramirez, Justin Savage, and Cagla Eroglu

Corresponding Author(s): Cagla Eroglu, Duke University School of Medicine

Review Timeline:

Submission Date:	2023-03-29
Editorial Decision:	2023-04-28
Revision Received:	2023-07-14
Editorial Decision:	2023-08-03
Revision Received:	2023-08-17

Monitoring Editor: Marc Freeman

Scientific Editor: Tim Spencer

Transaction Report:

DOI: <https://doi.org/10.1083/jcb.202303138>

April 28, 2023

Re: JCB manuscript #202303138

Dr. Cagla Eroglu
Duke Medical Center
Campus Box 3709
Durham, NC 27710

Dear Cagla,

Thank you for submitting your manuscript entitled " δ -catenin controls astrocyte morphogenesis via layer-specific astrocyte-neuron cadherin interactions". The manuscript was assessed by expert reviewers, whose comments are appended to this letter. We invite you to submit a revision if you can address the reviewers' key concerns, as outlined here.

You will see that all three reviewers are enthusiastic about the study but have raised a handful of concerns that we agree would help to support the main conclusions of the study and increase the impact of the work. In particular, the reviewers feel that you should clarify the expression patterns and timing of δ -cat in vivo (revs #1 & 3), quantify the knockdown better (rev#1), examine the effect of δ -cat knockdown in neuron-free cultures (including mosaic cultures) (rev#2), and look at neuronal morphology and synapses (all three reviewers). These issues should be addressed in the revised manuscript.

Reviewer #1 has also suggested that you look at Cdh10, 11, and 20 expression in vivo and test the sufficiency of δ -cat while reviewer #2 has suggested that you perform δ -cat/cadherin double knockdowns. We completely agree with the reviewers that performing these experiments would enhance the impact and broad appeal of the study and so would encourage you to address these issues in full, if feasible. However, they are not strictly necessary to support the main conclusions of the study and so they will not be required for resubmission - it's your call. However, please do provide responses to all of the reviewer points in your rebuttal document. We also hope that you will be able to address each of the other minor reviewer comments in full.

Finally, reviewer #1 has suggested that the structural diagrams in figure 3C-D and F are unnecessary - while we appreciate the reviewer's point, we feel that these diagrams will be useful for the non-specialist reader and so we would prefer to keep them in the paper.

GENERAL GUIDELINES:

Text limits: Character count for an Article is < 40,000, not including spaces. Count includes title page, abstract, introduction, results, discussion, and acknowledgments. Count does not include materials and methods, figure legends, references, tables, or supplemental legends.

Figures: Articles may have up to 10 main text figures. Figures must be prepared according to the policies outlined in our Instructions to Authors, under Data Presentation, <https://jcb.rupress.org/site/misc/ifora.xhtml>. All figures in accepted manuscripts will be screened prior to publication.

Supplemental information: There are strict limits on the allowable amount of supplemental data. Articles may have up to 5 supplemental figures. Up to 10 supplemental videos or flash animations are allowed. A summary of all supplemental material should appear at the end of the Materials and methods section.

Please note that JCB now requires authors to submit Source Data used to generate figures containing gels and Western blots with all revised manuscripts. This Source Data consists of fully uncropped and unprocessed images for each gel/blot displayed in the main and supplemental figures. Since your paper includes cropped gel and/or blot images, please be sure to provide one Source Data file for each figure that contains gels and/or blots along with your revised manuscript files. File names for Source Data figures should be alphanumeric without any spaces or special characters (i.e., SourceDataF#, where F# refers to the associated main figure number or SourceDataFS# for those associated with Supplementary figures). The lanes of the gels/blots should be labeled as they are in the associated figure, the place where cropping was applied should be marked (with a box), and molecular weight/size standards should be labeled wherever possible.

Source Data files will be made available to reviewers during evaluation of revised manuscripts and, if your paper is eventually

published in JCB, the files will be directly linked to specific figures in the published article.

The typical timeframe for revisions is three to four months. While most universities and institutes have reopened labs and allowed researchers to begin working at nearly pre-pandemic levels, we at JCB realize that the lingering effects of the COVID-19 pandemic may still be impacting some aspects of your work, including the acquisition of equipment and reagents. Therefore, if you anticipate any difficulties in meeting this aforementioned revision time limit, please contact us and we can work with you to find an appropriate time frame for resubmission. Please note that papers are generally considered through only one revision cycle, so any revised manuscript will likely be either accepted or rejected.

Thank you for this interesting contribution to Journal of Cell Biology. You can contact us at the journal office with any questions, cellbio@rockefeller.edu.

Sincerely,

Marc Freeman, PhD
Monitoring Editor
Journal of Cell Biology

Tim Spencer, PhD
Executive Editor
Journal of Cell Biology

Reviewer #1 (Comments to the Authors (Required)):

In this interesting paper by Tan, et al. the role of catenin/cadherin interactions in astrocyte morphogenesis is examined. They show that δ -catenin is expressed in developing cortical astrocytes and that its knockdown in vitro suppresses astrocyte morphogenesis, with astrocytes exhibiting a truncated morphology. Mechanistically, they show that δ -catenin interacts with cadherins, presumably from neurons and that δ -catenin mutants present in autism cannot interact with cadherins nor rescue morphogenesis in vitro. In vivo studies, using the PALE approach show that knockdown of δ -catenin impacts astrocyte morphogenesis in the cortex and that this effect is reliant upon cadherins in the deeper cortical layers. Overall, this is an interesting paper that extends the model that structural interactions between astrocyte and neurons are important for astrocyte morphogenesis and adds some new insights and mechanisms to this important process. However, there are some issues with the experimental design and some additional experiments that need to be conducted to fully convince this reviewer that this model is in fact what is occurring in vivo, during development. Accordingly, I hope that the authors will take these suggestions under advisement as they revise this potentially interesting paper:

- 1) In Figure 1, what percentage of astrocytes express δ -catenin? Quantification is needed.
- 2) Also, what is the timing of δ -catenin induction? Is it induced coincident with the gliogenic switch, or is it always expressed in radial glia or is it induced in differentiating astrocytes? The timing is really important here.
- 3) The shRNAi experiment is decent, but does overexpression of δ -catenin promote or accelerate astrocyte morphogenesis? This is an important experiment to distinguish between necessity and sufficiency. Also, this experiment should be done in vivo with PALE.
- 4) If they overexpress cadherin in upper layer neurons, does it promote astrocyte morphogenesis? This would be a great way to prove that they deep/upper layer effects are reliant upon cadherins.
- 5) Figures 3C-D, F seem excessive and unnecessary. Lines 200-209 spell out the logic very nicely, the figures don't really help.
- 6) The co-IPs are performed in 293 cells. All of these studies should be redone with astrocytes, as cell context matters and it's

important to show that these interactions occurs specifically in astrocytes. Similarly, the structure/function studies should also be done in astrocytes, as well as the surface biotin studies. While the R713C mutant seems to be important, its critical to do these structure/function studies in relevant cell types

7) They should show in vivo expression of Cdh10, 11, 20 in developing cortical astrocytes; similar to figure 1, with quantification.

8) The PALE experiments are nice, but the authors need to show knockdown of transfected cells in vivo and quantify the extent of knockdown---there is a lot of inherent variability with PALE between mice.

9) Moreover, the authors should repeat the structure/function studies with human δ -catenin and the R713 mutant, in vivo with PALE. This would be a nice complement to the in vitro studies.

10) What happens to neuronal synapses in the δ -catenin knockdown experiment? If astrocyte morphology is important for synapse formation and function, then manipulating δ -catenin (both overexpression and knockdown) should impact synapses. This would be important to show.

11) The final claim of this study is the layer-specific effects and a "combinatorial code". This is fine and somewhat interesting. They claim is this based on elevated expression of Cdh2/cadherin in deeper layer neurons-lines 381-382. To give the reader more confidence in this idea, they need to show via immunostaining or RNA-Scope or some other method that in fact Cdh2 is in fact more highly expressed in these deep layer neurons.

Reviewer #2 (Comments to the Authors (Required)):

In " δ -catenin controls astrocyte morphogenesis via layer-specific astrocyte-neuron cadherin interactions," Tan et al. explore how δ -catenin-cadherin, astrocyte-neuron interactions reciprocally regulate astrocyte morphogenesis. The authors used an elegant combination of in vitro and in vivo assays to demonstrate that loss of δ -catenin from astrocytes OR neurons significantly hampers astrocyte morphological maturation. Moreover, they show that the autism-associated δ -catenin mutation, R713C, disrupts δ -catenin-dependent surface localization of cadherins, and in turn, reduced astrocyte complexity. Finally, given the layer-specific expression pattern of different cadherins, the authors explore whether loss of N-cadherin (lower neurons only) would have a regional impact on astrocyte morphology. They show that loss of N-cadherin specifically disrupts astrocyte morphogenesis in lower layers of the cortex. These data lead the authors to posit that a layer-specific adhesion code might guide the development of unique astrocyte morphologies in the cortex.

This is a beautiful study from the Eroglu lab, who are leading experts in molecular regulation of astrocyte morphology. The data are very clear and thorough, and I have only a few suggestions to bolster their claims:

Major:

1. Throughout the manuscript, the authors claim that the effects of δ -catenin on astrocyte morphology are neuron contact-dependent, since astrocytes only gain an elaborate morphology in the presence of neurons. It would bolster this claim to perfect δ -catenin knockdown in an astrocyte monoculture to see if there are any changes in overall astrocyte shape.

2. Given that δ -catenin and Cadherins are expression in both neurons and astrocytes, and δ -catenin has a known role in dendrite formation, it would be interesting to see if knockdown of astrocyte δ -catenin alters the morphology of the neurons in their co-culture system. This is an important question, as changes in neuronal morphology and synaptogenesis would also impact astrocyte morphology.

3. As astrocytes tile with one another, and astrocytes express both δ -catenin and cadherins, it is possible that part of the observed changes in astrocyte morphology following δ -catenin knockdown are due to defective astrocyte-astrocyte adhesion. One way this could be addressed is by mixing wildtype astrocytes and δ -catenin knockdown astrocytes in culture and assessing nonautonomous effects.

4. Similarly, what is the effect of performing double knockdowns of δ -catenin and cadherin in astrocytes- does this enhance the morphogenesis defect, or are they epistatic to one another?

Minor:

1. The authors should improve labeling throughout their figures to indicate the developmental stage (postnatal day or day in vitro) of the data shown (e.g. Figure 1 C-D, Figure 2, Figure 4, Figure 6).

2. It would be helpful to the reader to add labels to indicate where the human mutations are within the cartoon of δ -catenin (Figure 3a).

Reviewer #3 (Comments to the Authors (Required)):

The current manuscript entitled 'δ-catenin controls astrocyte morphogenesis via layer-specific astrocyte-neuron cadherin interactions' aims to identify δ-catenin as a critical protein in astrocyte morphogenesis. The work details a novel mechanism in mouse cortex development by which the cadherin-astrocyte catenin complex contributes to astrocyte morphology, specifically in deeper cortical layers where neuronal cadherin expression is high. This work is interesting as it starts to address neuronal specific cues that may function to drive astrocyte morphogenesis and details an exciting role for δ-catenin in astrocyte maturation. The authors provide a computational model system to examine the potential binding domains that regulate the δ-catenin/cadherin interactions, and used IP to test this in vitro. Using short-hairpin RNA knockdown, in vitro and in vivo, the authors also demonstrated that knockdown of δ-catenin in astrocytes stunts astrocyte morphology, knock down of neuronal catenin inhibits astrocyte morphogenesis and knock down of astrocyte cadherin specifically alters astrocyte morphology in lower level astrocytes. Generally, this paper is of broad interest to neurodevelopmental and glial biologists. Suggested points of improvement and clarification are indicated below.

Major

A useful piece of data would be to show δ-catenin levels across the different layers of the cortex in astrocytes (and neurons). This data should be available based on figure 1C,D (RNA probes or immuno).

The differences seen in astrocyte morphology in the upper layers when δ-catenin is disrupted in neurons may be secondary to disrupted neuronal maturation. Given the importance of δ-catenin in neuronal development morphology/synapse # should be quantified. To this figure it would also be useful to add an additional panel demonstrating efficiency of δ-catenin KO in upper levels neurons.

Similarly, it would also be useful to demonstrate the cadherin KO in astrocytes in figure 7.

The authors demonstrate overall astrocyte branch length is unaffected while complexity is affected. Is layer thickness affected in animal perturbations (specifically at the earlier timepoint when the differences in astrocyte morphology are so profound)?

In culture experiments, the authors use a 48-hour time point (astrocytes cultured with neurons) to assess astrocyte complexity, while in vivo they show that over time (by P21) astrocyte volume catches up to WT. Please briefly discuss why the 48-hour time point was chosen and did the authors evaluate a later (i.e. 7 day time point)?

Please clarify what constitutes upper layers and lower layers (where does layer IV fall).

Layer 4 excitatory neurons are quite different in morphology from layer 2/3 and 5/6 (spiny rather than pyramidal). For the reader it would be useful to discuss astrocyte morphology across the layers

It is not clear which layers astrocytes were quantified for in figure 5. Was this across all layers, if so, results should be binned by layer (layer 2/3, layer 4 and layer 5/6 - minimum 3 bins) to demonstrate equivalent impact of knockdown across layers.

Minor:

Figure 1 A could be removed. It does not add to the manuscript and is covered in detail in other articles, consider just citing.

The manuscript is dense and required back and forth between the text, figure legends and the figures to determine cadherin and δ-catenin manipulations in different cells type and layers. There also seems to be extraneous text. While I have no specific recommendations to consolidate, consider editing to streamline. A concluding cartoon may also be useful.

We thank the reviewers for carefully reviewing and insightful critiquing our manuscript, “ δ -catenin controls astrocyte morphogenesis via layer-specific astrocyte-neuron cadherin interactions.” We were pleased to see that all three reviewers found this manuscript provided novel insights, is interesting to a broad audience, and acknowledged the experimental rigor and thoroughness of the study. However, the reviewers also identified some concerns and suggested experiments to improve the study. Below we first respond to the concerns highlighted by the editor, which we addressed by performing the experiments detailed below. This response is followed by our point-by-point response to reviewer comments which we addressed fully through new experiments or textual changes and clarifications. The results of the new experiments and analyses, together with suggested edits in figures, now fully address all the major concerns raised by the reviewers and improve our manuscript immensely.

Point-by-point response:

Editor concern 1: Clarify the expression patterns and timing of δ -catenin *in vivo*.

This is in line with comments from reviewer1 “What percentage of astrocytes express δ -catenin? Also, what is the timing of δ -catenin induction (and) show via immunostaining or RNAscope that *Cdh2* is more highly expressed in deep layer neurons” and reviewer 3 “A useful piece of data would be to show δ -catenin levels across the different layers of the cortex in astrocytes (and neurons).”

We clarified the expression patterns and timing of δ -catenin *in vivo* by quantifying astrocytic *Ctnnd2* mRNA expression in the different cortical layers at P1, P7, P14, and P21 through multiplexed quantitative high-resolution immunohistochemistry and RNA fluorescence in situ hybridization (FISH). This allowed us to quantify *Ctnnd2* mRNA expression within the soma of GFP-labeled astrocytes in the primary visual cortex as described in Farhy-Tselnicker et al., 2021.

In total, we quantified *Ctnnd2* mRNA expression in an average of 83 astrocytes/layer/timepoint, normalizing the volume of *Ctnnd2* mRNA expression to the volume of astrocyte soma. We found that *Ctnnd2* mRNA levels did not vary significantly across cortical layers at P1, P7, P14 and P21 (Fig 1e). However, comparing *Ctnnd2* mRNA expression of each layer across the first 3 postnatal weeks revealed that *Ctnnd2* mRNA expression in layers 1, 2/3, 4, and 5 was significantly different across this critical developmental time period (Fig 1f). It should also be noted that our quantitative RNA-FISH experiments also revealed that *Ctnnd2* mRNA expression decreases significantly by P21, congruent with our previous quantification of δ -catenin expression *in vivo* (Fig 1b).

Editor concern 2: Quantify the knockdown better. This addresses the concerns of reviewer 1 “The authors need to show knockdown of transfected cells *in vivo* and quantify the extent of knockdown---there is a lot of inherent variability with PALE between mice.”

We performed immunohistochemistry stains of δ -catenin (Abcam 184917) on 40 μ m sections of sh*Ctnnd2* and scr*Ctnnd2* PALE brains. Specifically, we quantified the volume

of δ -catenin/Ctnnd2+ puncta within a region of interest (ROI), normalized to the volume of that ROI. 3 ROIs were quantified per astrocyte, and a total of 18 astrocytes from 5-7 brains were imaged and analyzed per condition. In PALE astrocytes, there was a significant (41% reduction) in δ -catenin expression ($p=0.036$, Fig S4a and S4b).

Editor concern 3: Examine the effect of δ -catenin knockdown in neuron-free cultures and mosaic astrocyte cultures. This addresses two concerns from reviewer 2 described below:

“Throughout the manuscript, the authors claim that the effects of δ -catenin on astrocyte morphology are neuron contact-dependent since astrocytes only gain an elaborate morphology in the presence of neurons. It would bolster this claim to perform δ -catenin knockdown in an astrocyte monoculture to see if there are any changes in overall astrocyte shape.”

“As astrocytes tile with one another, and astrocytes express both δ -catenin and cadherins, it is possible that part of the observed changes in astrocyte morphology following δ -catenin knockdown are due to defective astrocyte-astrocyte adhesion. One way this could be addressed is by mixing wildtype astrocytes and δ -catenin knockdown astrocytes in culture and assessing nonautonomous effects.”

We answered these two important questions with the following experiments:

First, we characterized astrocyte morphology in astrocyte monocultures in a culture system closely mimicking our astrocyte-neuron coculture system (Fig 2a). At DIV11, 64K wildtype astrocytes were cultured with 16K shControl-GFP or shCtnnd2-GFP transfected astrocytes for 48h in serum-free media.

As expected, both shControl and shCtnnd2 astrocytes in the astrocyte monoculture were simple, as evidenced by the significantly reduced complexity in astrocyte processes when compared to shControl astrocytes co-cultured with neurons ($p < 2.2 \times 10^{-16}$ both conditions, Fig 2b and 2c). More critically, shControl and shCtnnd2 monoculture astrocytes were indistinguishable from each other in terms of morphology ($p=0.09$), and both shCtnnd2 monoculture astrocytes and shCtnnd2 astrocytes co-cultured with neurons had similar complexities ($p=0.97$) when quantified by Sholl analysis (Fig 2b and 2c). Taken together, these experiments provided strong evidence that the effects of δ -catenin loss on astrocyte morphology is neuronal contact-dependent.

To test if observed changes in astrocyte morphology following δ -catenin knockdown can also be impacted by defective astrocyte-astrocyte adhesion, we cultured astrocytes in the following paradigm we term astrocyte mosaic culture (Fig S2e). Briefly, 1.5-2million astrocytes/condition were nucleoporated with the following constructs at DIV 7:

1. scrCtnnd2-mCherry
2. shCtnnd2-mCherry
3. scrCtnnd2-GFP
4. shCtnnd2-GFP

Nucleoporation was utilized as it afforded higher transfection efficiency (~80% for mCherry constructs and ~90% for GFP constructs) compared to the previously described transfection protocol (~20% efficiency). The nucleoporated astrocytes were co-cultured in the following manner with a total density of 80K astrocytes/coverlip. After a 12-16h recovery period from nucleoporation, the astrocytes were cultured in the same neuronal growth media used in other experiments for 48h before fixation.

The conditions were as follows

1. 90% scrCtnnd2-mCherry astrocytes with 10% scrCtnnd2-GFP astrocytes.
2. 90% scrCtnnd2-mCherry astrocytes with 10% shCtnnd2-GFP astrocytes. This will test the cell-autonomous effect of Ctnnd2 on astrocyte morphology independent of neuron contact
3. 90% shCtnnd2-mCherry astrocytes with 10% scrCtnnd2-GFP. This will test the cell non-autonomous effect of Ctnnd2 on astrocyte morphology independent of neuron contact
4. 90% shCtnnd2-mCherry astrocytes with 10% shCtnnd2-GFP. This is the negative control for this experiment

Only single GFP+ astrocytes, which were fully surrounded by the tdTomato+ astrocytes were imaged and assessed for astrocyte morphology.

As expected, we found that astrocytes in the mosaic monoculture paradigm were less complex than astrocytes in the neuron co-culture system, with a peak of only 8-10 intersections when quantified by Sholl analysis across all conditions tested (Fig S2f and S2g). As a control, we first compared the morphology of shControl astrocytes in shControl astrocyte mosaic culture with shControl astrocytes in shCtnnd2 astrocytes mosaic culture. As with the astrocyte monoculture paradigm, astrocytes grown without neurons had no complex morphology, and silencing Ctnnd2 did not influence wildtype astrocyte morphology ($p=0.18$). Next, we determined that wildtype astrocyte morphology was unchanged when cultured in either shControl or shCtnnd2 astrocytes ($p=0.06$). The same was also observed in shCtnnd2 astrocytes cultured in shControl or shCtnnd2 astrocytes ($p=0.91$). Taken together, these two experiments reveal that the observed changes in astrocyte morphology following δ -catenin knockdown in astrocyte-neuron co-cultures are not due to defective astrocyte-astrocyte adhesion.

Editor concern 4: Examine the effect of δ -catenin knockdown on neuronal morphology and synapses. This critique is common to all reviewers, and we agree that these suggested experiments are vital controls that would boost rigor of our study.

Reviewer 1: "What happens to neuronal synapses in the δ -catenin knockdown experiment? If astrocyte morphology is important for synapse formation and function, then manipulating δ -catenin (both overexpression and knockdown) should impact synapses. This would be important to show."

Reviewer 2: "Given that δ -catenin and Cadherins are expressed in both neurons and astrocytes, and δ -catenin has a known role in dendrite formation, it would be interesting to see if knockdown

of astrocyte δ -catenin alters the morphology of the neurons in their co-culture system. This is an important question as changes in neuronal morphology, and synaptogenesis would also impact astrocyte morphology.”

Reviewer 3: “The differences seen in astrocyte morphology in the upper layers when δ -catenin is disrupted in neurons may be secondary to disrupted neuronal maturation. Given the importance of δ -catenin in neuronal development, morphology/synapse number should be quantified.”

We addressed these questions with two new experiments. First, to determine if astrocytic Ctnnd2 have non-cell autonomous effects on neuronal morphology, we developed a new co-culture assay in which we manipulated the astrocytes but assessed neuronal outgrowth. To do so, we plated 5K neurons isolated from P1 rat cortices onto a monolayer of DIV 7 astrocytes nucleoporated with either shCtnnd2-GFP or the scrambled control (scrCtnnd2-GFP). Neurons were co-cultured with astrocytes for 5 days in serum-free media to allow for neurite growth and stained with Tuj1 to assess neuron morphology (Fig S2b). Our analysis revealed that neuron morphology was unaffected by the silencing of astrocytic Ctnnd2 *in vitro* ($p= 0.58$, Fig S2c and S2d). These results suggest that astrocytic Ctnnd2 does not have direct impact on neuronal morphology.

To assess the effect of astrocytic Ctnnd2 on synapses, we quantified the number of synapses in P14 shControl and shCtnnd2 PALE astrocytes (Fig S4e). P14 was chosen as the timepoint for two reasons: endogenous δ -catenin expression peaks at P14, and we observed the strongest difference in astrocyte complexity and territory volume at this time point. We did not observe a significant change in the density of excitatory synapses within the territories of astrocytes ($p=0.73$, Fig S4f and S4g) within the territory of astrocytes in which Ctnnd2 was silenced compared to controls.

These new results indicate that while neuronal Ctnnd2 is essential for astrocytic morphogenesis, astrocytic Ctnnd2 may be dispensable for neuronal morphogenesis and excitatory synapse formation. This possibility is also in line with our model that the neurons, which are born first, establish the cadherin code within cortical layers, and astrocytes that arrive later utilize this code to direct their morphogenesis. We now discuss this in the manuscript.

In the next section, we will give a point-by-point response to the individual points raised by each reviewer:

Reviewer 1 point 1: In vivo rescue of astrocyte morphogenesis by PALE. Specifically, reviewer 1 would like to know if “overexpression of δ -catenin promotes or accelerates in astrocyte morphogenesis (..) in vivo with PALE” and requested that we “repeat the structure/function studies with human δ -catenin and R713C mutant in vivo with PALE”

In our paper, we show that overexpression of δ -catenin does not further increase astrocyte morphological complexity in vitro (Fig 2d-f). There was no significant difference in astrocyte complexity or total branch length when comparing wildtype astrocytes (shControl) with astrocytes overexpressing δ -catenin (shControl + hCTNND2-HA) *in vitro*. We agree with Reviewer 1 that this observation would be strengthened if we conducted *in vivo* rescue experiments since astrocytes are more complex *in vivo* than *in vitro*.

Unfortunately, our attempts at these experiments proved unsuccessful. Overexpression of δ -catenin at late P0 (shControl-mCherryCAAX + hCTNND2-HA) using our PALE paradigm did not yield any double-labelled astrocytes (Response Fig 1). We decreased the concentration of hCTNND2-HA construct for observing any successful labeling of astrocytes. These observations suggested that overexpression of δ -catenin in radial glial cells (RGCs) by PALE is likely detrimental for astrocytogenesis.

Response Figure 1. Introduction of shControl-mCherryCAAX and hCTNND2-HA constructs at late P0 using PALE paradigm did not yield any astrocytes overexpressing δ -catenin by P14.

When we overexpressed hCTNND2-HA in δ -catenin knockdown astrocytes using PALE (shCtnnd2-mCherryCAAX + hCTNND2-HA), we again did not observe any double-labelled astrocytes. However, we noted the presence of double-labeled RGCs at the sub-ventricular zone (Response Fig 2). The processes of these double-labeled radial glial

cells were unable to extend outside of the SVZ (Fig B), indicating that overexpression of Ctnnd2 in RGCs by PALE leads to deficits in RGC morphology and function. Taken together, these findings underscore the need for Ctnnd2 expression to be tightly regulated in RGCs during early cortical development for normal astrogenesis. Also, these results show that we cannot perform in vivo rescue experiment by PALE.

Response Figure 2. Radial glial cells transfected with shCtnnd2-mCherryCAAX and hCTNND2-HA were unable to extend their processes beyond the subventricular zone.

Reviewer 1 point 2: Structure/function analyses in astrocyte cultures. Reviewer 1 suggested that the biochemical experiments described in Fig 3 be “redone with astrocytes, as cell context matters and it is important to show that these interactions occur specifically in astrocytes”.

While we agree with Reviewer 1 that cell context matters, this is not an experiment that is currently technically feasible. There are no good antibodies that can be used for immunoprecipitating these two proteins. Importantly, the interaction between cadherins and δ -catenin has been found to be robust across cell types (Kim et al., 2008; Van Hoorde et al., 2009).

However, we performed another experiment in astrocytes to functionally link surface expression of N-cadherin to δ -catenin. To do so, we nucleoporated shControl-GFP or shCtnnd2-GFP into DIV7 astrocytes and seeded them at 300K/well. These astrocytes were cultured for another 4 days (to reach confluency) before surface biotinylation assays, as previously described in the manuscript. We quantified the expression of cell-surface N-cadherin in shControl-GFP (wildtype) and shCtnnd2-GFP (δ -catenin knockdown) astrocytes.

In this paradigm, there was a $52.5 \pm 0.03\%$ reduction ($p=0.004$) in δ -catenin expression in shCtnnd2 astrocyte lysates compared to shControl (Fig 4d and Fig S3a). The total N-cadherin expression was not altered ($p=0.13$, Fig 4d and Fig S3b). However, there was a significant and striking $62.4 \pm 0.1\%$ decrease in surface expression of endogenous N-cadherin in shCtnnd2 compared to shControl astrocytes ($p=0.023$, Fig 4d and 4e). The results of this experiment in astrocytes are in line with the conclusions we drew from the biochemical assays performed in HEK293T cultures.

Reviewer 2 point 1: Epistasis of δ -catenin and cadherin in astrocyte. Reviewer 2 would like to know the “effect of performing double knockdowns of δ -catenin and cadherin in astrocytes” and if it “enhance(s) the morphogenesis defect, or epistatic to one another”.

We agree with Reviewer 2 that this is an important experiment and conducted shCtnnd2/shCdh2 double knockdown assays *in vitro*. Both populations of astrocytes had strongly diminished astrocyte complexity similar to shCtnnd2 astrocytes, indicative of epistasis between δ -catenin and cadherin in astrocytes (Fig 4f). We found no statistical difference between scrCdh2 + shCtnnd2 and shCdh2 + shCtnnd2 astrocyte morphology ($p=0.57$, Fig 4g). Furthermore, we also noted that Ctnnd2 and Cdh2 double knockdown astrocytes only had a few short primary processes which is characteristic of shCtnnd2 astrocytes (Fig 4f). Comparison of both conditions by Sholl analysis confirm that knockdown of both Ctnnd2 and Cdh2 did not enhance the morphogenesis defect ($p=0.70$, Fig 4g) and is indicative that Ctnnd2 and Cdh2 work in the same molecular pathway to regulate astrocyte morphogenesis.

Reviewer 2 point 2: Clarity in figure labeling. Reviewer 2 commented that labeling throughout our figures be improved to “indicate developmental stage” and “where human mutations are within the cartoon of δ -catenin (Figure 3a)”.

We thank Reviewer 2 for this feedback and have included more specific labels in all figures as suggested for improved clarity, taking care to include developmental stages and also rectifying Fig 3a.

Reviewer 3 point 1: Effect of δ -catenin on overall brain morphology. Reviewer 3 questions if “(cortical) layer thickness (is) affected in animal perturbations, specifically at the earlier timepoints when the differences in astrocyte morphology are so profound”.

Because PALE only labels a sparse population of astrocytes, it is unlikely to affect overall brain morphology. However, for scientific rigor, we analyzed the cortical thickness (indicated by blue arrow, Response Figure 3a) of P14 shControl and shCtnnd2 PALE cortex as well as the ratio of upper layer cortex (Satb2+ cells) relative to total cortex thickness (indicated by orange arrow, Response Figure 3a). We found no significant changes in both measures. Cortical thickness in shControl and shCtnnd2 PALE brains varied by 0.54% (p value =0.86, Response Figure 3b) and the thickness of upper layer cortex was approximately 53% of total cortical thickness in shControl and shCtnnd2 PALE brains ($p=0.24$, Response Figure 3c).

Response Figure 3. Overall brain morphology is unaffected by silencing δ -catenin in a sparse population of astrocytes by PALE.

Reviewer 3 point 2: Rationale for 48h co-culture. Reviewer 3 would like us to briefly discuss why the 48h timepoint was chosen and if (we) evaluated a later timepoint.

The suitable timepoints for cortical astrocyte-neuron co-culture were extensively characterized in a prior Eroglu lab publication authored by Stogsdill et al. (2017, Extended data Fig 2). Specifically, the timepoints 6h, 12h, 24h, 48h, 72h and 120hr were evaluated for astrocyte complexity in astrocyte monoculture and astrocyte-neuron co-culture. We found that astrocyte complexity in co-culture conditions does not vary much between 24h to 120h timepoint. 48h was chosen as a standard in the lab because tripartite synapses were detected at this timepoint. We now mention this in the materials and methods.

Reviewer 3 point 3: Clarification of PALE quantification by cortical layer. Reviewer 3 wanted clarification on “what constitutes upper and lower layers” and requested for quantification of Fig 5 by layers to determine if there was an “equivalent impact of knockdown across layers.”

In our experiments, upper layer includes layer 1 to layer 4 (layer 2-4 as Satb2 positive) while lower layer comprises of layer 5 and 6 (Ctip2 positive). Representative images of the shControl and shCtnnd2 PALE brains can be found in Response Fig S3. Satb2 and Ctip2 are complementary markers for upper-layer and deep-layer pyramidal neurons in the developing cortex (Britanova et al., 2008, Neuron). We agree that the quantification of Fig 5 by upper- and lower-layers might be informative and have added this as Fig S4c

and S4d for P14 PALE astrocytes. P14 is when δ -catenin expression peaks *in vivo* and when the reduction in astrocyte complexity is most severe.

August 3, 2023

RE: JCB Manuscript #202303138R

Dr. Cagla Eroglu
Duke University School of Medicine
Department of Cell Biology
333A Nanaline Duke Building
307 Research Drive
Durham, North Carolina 27710

Dear Cagla:

Thank you for submitting your revised manuscript entitled " δ -catenin controls astrocyte morphogenesis via layer-specific astrocyte-neuron cadherin interactions". The paper has now been seen again by the original reviewers, all of whom now recommend acceptance. Therefore, we would be happy to publish your paper in JCB pending final revisions necessary to meet our formatting guidelines (see details below).

A. MANUSCRIPT ORGANIZATION AND FORMATTING:

1) Text limits: Character count for Articles is < 40,000, not including spaces. Count includes the abstract, introduction, results, discussion, and acknowledgments. Count does not include title page, materials and methods, figure legends, references, tables, or supplemental legends. While you are over the limit, we should be able to give you the extra space this time. However, please try not to add substantively to the character count in revision.

2) Figure formatting: Scale bars must be present on all microscopy images, including inset magnifications. Molecular weight or nucleic acid size markers must be included on all gel electrophoresis.

****Please add molecular weight markers to the blots in Figure 1b and Supplementary figure 2a and 2h.****

3) Statistical analysis: Error bars on graphic representations of numerical data must be clearly described in the figure legend. The number of independent data points (n) represented in a graph must be indicated in the legend. Statistical methods should be explained in full in the materials and methods. For figures presenting pooled data the statistical measure should be defined in the figure legends. Please also be sure to indicate the statistical tests used in each of your experiments (both in the figure legend itself and in a separate methods section) as well as the parameters of the test (for example, if you ran a t-test, please indicate if it was one- or two-sided, etc.).

****Also, since you used parametric tests in your study (e.g. t-tests, ANOVA, etc.), you should have first determined whether the data was normally distributed before selecting that test. In the stats section of the methods, please indicate how you tested for normality. If you did not test for normality, you must state something to the effect that "Data distribution was assumed to be normal but this was not formally tested."****

4) Materials and methods: Should be comprehensive and not simply reference a previous publication for details on how an experiment was performed. Please provide full descriptions (at least in brief) in the text for readers who may not have access to referenced manuscripts. The text should not refer to methods "...as previously described."

5) Please be sure to provide the sequences for all of your primers/oligos and RNAi constructs in the materials and methods. You must also indicate in the methods the source, species, and catalog numbers (where appropriate) for all of your antibodies.

6) Microscope image acquisition: The following information must be provided about the acquisition and processing of images:

- a. Make and model of microscope
- b. Type, magnification, and numerical aperture of the objective lenses
- c. Temperature
- d. imaging medium
- e. Fluorochromes
- f. Camera make and model
- g. Acquisition software

h. Any software used for image processing subsequent to data acquisition. Please include details and types of operations involved (e.g., type of deconvolution, 3D reconstitutions, surface or volume rendering, gamma adjustments, etc.).

7) References: There is no limit to the number of references cited in a manuscript. References should be cited parenthetically in the text by author and year of publication. Abbreviate the names of journals according to PubMed.

8) Supplemental materials: There are strict limits on the allowable amount of supplemental data. Articles may have up to 5 supplemental figures. At the moment, you currently meet this limit but please bear it in mind when revising. Please also note that tables, like figures, should be provided as individual, editable files. A summary of all supplemental material (that is, in addition to the supplementary figure legends) should appear at the end of the Materials and methods section. Please see any recent JCB paper for an example of this.]

9) eTOC summary: A ~40-50 word summary that describes the context and significance of the findings for a general readership should be included on the title page.

****While we realize that you have already provided one, the statement should be written in the present tense and refer to the work in the third person. It should contain "First author name(s) et al..." to match our preferred style.****

10) Conflict of interest statement: JCB requires inclusion of a statement in the acknowledgements regarding competing financial interests. If no competing financial interests exist, please include the following statement: "The authors declare no competing financial interests." If competing interests are declared, please follow your statement of these competing interests with the following statement: "The authors declare no further competing financial interests."

11) A separate author contribution section is required following the Acknowledgments in all research manuscripts. All authors should be mentioned and designated by their first and middle initials and full surnames. We encourage use of the CRediT nomenclature (<https://casrai.org/credit/>).

12) ORCID IDs: ORCID IDs are unique identifiers allowing researchers to create a record of their various scholarly contributions in a single place. Please note that ORCID IDs are now ***required*** for all authors. At resubmission of your final files, please be sure to provide your ORCID ID and those of all co-authors.

13) As you know, JCB now requires authors to submit Source Data used to generate figures containing gels and Western blots with all revised manuscripts. Thank you for providing the Source Data for your gel/blot images. However, Source Data figures should be provided as individual PDF files (****one file per figure****). Specifically, please put all three blots for main figure 3 into a single (multipage) file and both blots from supp figure 2 into a single (multipage) file. Authors should endeavor to retain a minimum resolution of 300 dpi or pixels per inch. Please review our instructions for export from Photoshop, Illustrator, and PowerPoint here: <https://rupress.org/jcb/pages/submission-guidelines#revised>

14) Journal of Cell Biology now requires a data availability statement for all research article submissions. These statements will be published in the article directly above the Acknowledgments. The statement should address all data underlying the research presented in the manuscript. Please visit the JCB instructions for authors for guidelines and examples of statements at (<https://rupress.org/jcb/pages/editorial-policies#data-availability-statement>).

B. FINAL FILES:

****It is JCB policy that if requested, original data images must be made available to the editors. Failure to provide original images upon request will result in unavoidable delays in publication. Please ensure that you have access to all original data images prior to final submission.****

****The license to publish form must be signed before your manuscript can be sent to production. A link to the electronic license to publish form will be sent to the corresponding author only. Please take a moment to check your funder requirements before choosing the appropriate license.****

Thank you for your attention to these final processing requirements. Please revise and format the manuscript and upload materials within 7-14 days. If complications arising from measures taken to prevent the spread of COVID-19 will prevent you from meeting this deadline (e.g. if you cannot retrieve necessary files from your laboratory, etc.), please let us know and we can work with you to determine a suitable revision period.

Please contact the journal office with any questions, cellbio@rockefeller.edu.

Thank you for this interesting contribution, we look forward to publishing your paper in Journal of Cell Biology.

Sincerely,

Marc Freeman, PhD
Monitoring Editor
Journal of Cell Biology

Tim Spencer, PhD
Executive Editor
Journal of Cell Biology

Reviewer #1 (Comments to the Authors (Required)):

The authors have adequately addressed my comments.
The paper is suitable for publication.

Reviewer #2 (Comments to the Authors (Required)):

In "δ-catenin controls astrocyte morphogenesis via layer-specific astrocyte-neuron cadherin interactions," Tan et al. explore how δ-catenin-cadherin, astrocyte-neuron interactions reciprocally regulate astrocyte morphogenesis. The authors used an elegant combination of in vitro and in vivo assays to demonstrate that loss of δ-catenin from astrocytes OR neurons significantly hampers astrocyte morphological maturation. Moreover, they show that the autism-associated δ-catenin mutation, R713C, disrupts δ-catenin-dependent surface localization of cadherins, and in turn, reduced astrocyte complexity. Finally, given the layer-specific expression pattern of different cadherins, the authors explore whether loss of N-cadherin (lower neurons only) would have a regional impact on astrocyte morphology. They show that loss of N-cadherin specifically disrupts astrocyte morphogenesis in lower layers of the cortex. These data lead the authors to posit that a layer-specific adhesion code might guide the development of unique astrocyte morphologies in the cortex.

This is a beautiful study from the Eroglu lab, who are leading experts in molecular regulation of astrocyte morphology. The authors did an excellent job addressing the concerns of each of the reviewers, and it is now ready for publication.

Reviewer #3 (Comments to the Authors (Required)):

The authors have done a nice job addressing comments. I have no additional concerns.